# Multi-Modal Image Registration Based on Phase Exponent Differences of the Gaussian Pyramid

Xiaohu Yan [1,2] , Yihang Cao [3], Yijun Yang [4] and Yongxiang Yao [3,*]

1   School of Undergraduate Education, Shenzhen Polytechnic University, Shenzhen 518055, China;
    yanxiaohu@szpt.edu.cn
2   Institute of Applied Artificial Intelligence of the Guangdong-Hong Kong-Macao Greater Bay Area,
    Shenzhen Polytechnic University, Shenzhen 518055, China
3   School of Remote Sensing and Information Engineering, Wuhan University, Wuhan 430079, China;
    2021302131083@whu.edu.cn
4   School of Artificial Intelligence, Shenzhen Polytechnic University, Shenzhen 518055, China;
    yangyijun2@szpt.edu.cn
*   Correspondence: yaoyongxiang@whu.edu.cn

**Abstract:** In multi-modal images (MMI), the differences in their imaging mechanisms lead to large signal-to-noise ratio differences, which means that the matching of geometric invariance and the matching accuracy of the matching algorithms often cannot be balanced. Therefore, how to weaken the signal-to-noise interference of MMI, maintain good scale and rotation invariance, and obtain high-precision matching correspondences becomes a challenge for multimodal remote sensing image matching. Based on this, a lightweight MMI alignment of the phase exponent of the differences in the Gaussian pyramid (PEDoG) is proposed, which takes into account the phase exponent differences of the Gaussian pyramid with normalized filtration, i.e., it achieves the high-precision identification of matching correspondences points while maintaining the geometric invariance of multi-modal matching. The proposed PEDoG method consists of three main parts, introducing the phase consistency model into the differential Gaussian pyramid to construct a new phase index. Then, three types of MMI (multi-temporal image, infrared–optical image, and map–optical image) are selected as the experimental datasets and compared with the advanced matching methods, and the results show that the NCM (number of correct matches) of the PEDoG method displays a minimum improvement of 3.3 times compared with the other methods, and the average RMSE (root mean square error) is 1.69 pixels, which is the lowest value among all the matching methods. Finally, the alignment results of the image are shown in the tessellated mosaic mode, which shows that the feature edges of the image are connected consistently without interlacing and artifacts. It can be seen that the proposed PEDoG method can realize high-precision alignment while taking geometric invariance into account.

**Keywords:** multi-modal images; phase exponent differences of gaussian pyramid; normalized filtering; log-polar coordinate descriptor; image registration

## 1. Introduction

With the rapid development of computer vision and multi-sensor technology [1] in recent years, the vast amount of multi-modal images has not only provided rich information but also poses significant challenges for image analysis [2]. Image registration is one of the fundamental tasks in the analysis of multi-modal images, aimed at geometrically aligning two or more images with overlapping regions. Multi-modal image (MMI) registration is an indispensable step in processes such as image fusion, image stitching, and change detection, and it has become a hot topic in international academic research [3].

The significant differences in illumination, contrast, geometric distortions, and non-linear radiometric variations in MMIs make them more sensitive to image gradients and

orientations, making it challenging to extract reliable common features and often resulting in subpar feature matching results. When registering MMIs, traditional parameter optimization methods can easily become trapped in local optima, leading to lower registration accuracy. In recent years, researchers have conducted extensive studies on MMI matching [4], continuously improving algorithms and evaluating accuracy to enhance the precision and reliability of registration. Despite substantial efforts to enhance the robustness of multi-modal image matching, challenges still persist in achieving both geometric invariance and high-precision registration.

To address the limitations of feature matching, as well as challenges posed by issues such as signal-to-noise ratio interference and low matching efficiency in multi-modal images. Conventional feature-based image registration methods like SIFT (the scale invariant feature transform) [5] and SURF (speeded-up robust features) [6] typically rely on gradient information to extract and describe features, which makes them sensitive to radiometric differences and unsuitable for multi-modal image registration. In contrast to gradients, image phase congruency (PC) features reflect structural information in images and are insensitive to radiometric variations [7]. Therefore, this paper combines the phase congruency model with differences in the Gaussian image pyramid and optimizes the SIFT feature detector along with normalized logarithmic polar descriptors, collectively referred to as the PEDoG algorithm. This algorithm captures more robust feature points, thereby improving the efficiency of multi-modal image matching while suppressing image noise and maintaining strong geometric invariance, facilitating robust matching in multi-modal images.

The main contributions of this study are as follows:

(1) A lightweight multi-modal image registration method was introduced that considers phase-indexed difference pyramids and normalized filtering. This approach achieves the high-precision recognition of corresponding points while maintaining geometric invariance in multi-modal matching.

(2) A method for constructing phase-indexed difference pyramids was proposed. By combining the phase congruency model with differences in the Gaussian pyramid and optimizing it using exponential functions, we established the phase-indexed difference pyramid image space. We also employed an SIFT-like feature extractor to extract robust feature points.

(3) A method for normalized filtering in logarithmic polar descriptors was introduced. This involves incorporating normalized filtering functions to enhance structural information in images, constructing second-order gradient-oriented features, and ultimately using a logarithmic polar coordinate framework to generate efficient descriptors that represent features in multi-modal images effectively.

The structure of this paper is as follows: In Section 1, we introduce the significance of multi-modal image matching and the importance of this research. Section 2 briefly reviews the limitations of previous studies. Section 3 outlines the complete process of the proposed PEDoG method. In Section 4, we provide a detailed analysis of the experiments conducted. Section 5 discusses the impact of different parameter settings on the performance of PEDoG and its performance concerning rotation and scale variations. Finally, in Section 6, we summarize the contributions of this study.

## 2. Relate Work

Image registration is one of the critical preprocessing steps in practical applications of remote sensing imagery and holds significant practical value. Its outcomes directly impact the accuracy of tasks such as image stitching, image fusion, and object detection. Image registration methods can be categorized into three main types: region-based registration methods, feature-based registration methods, and deep learning-based methods. The following section provides a detailed overview of each of these approaches.

Region-based registration methods establish registration relationships between image pairs using the raw pixel values and specific similarity metrics. These methods can be broadly categorized into three classes, including correlation-based methods [8], Fourier-

based methods [9], and mutual information-based methods [10]. In recent years, some researchers have introduced phase correlation techniques, leading to rapid advancements in multi-modal image matching. Building upon this, experts and scholars have developed methods that combine phase congruency models, self-similarity features, and enhanced gradient features. These include techniques such as directional gradient distance histograms combined with grey wolf optimization [11], log–Gabor filter-optimized matching HOPC (histogram of orientated phase congruency) [12], CFOG (channel features of orientated gradients) [13], angle-weighted orientation gradient (AWOG) [14], and multi-orientation tensor index feature (MoTIF) [15]. These approaches effectively overcome nonlinear radiometric distortions and contrast differences between multi-modal images and exhibit strong robustness in handling displacement variations in images. However, it is worth noting that such methods primarily focus on translational shifts, and if images involve complex geometric transformations, registration methods may encounter challenges.

Feature-based methods in the field of image registration began with Lowe et al.'s introduction of scale invariant feature transform (SIFT) matching [5], leading to the rapid development of various SIFT-like techniques [16]. However, gradient features are unable to adapt to the modality differences in multi-modal images, making such methods unsuitable for multi-modal image matching. Ma et al. introduced the PSO-SIFT (position scale orientation—SIFT) algorithm [17], which performs well in handling nonlinear brightness differences and rotation changes through the establishment of new image gradient features, but it remains sensitive to contrast differences and signal-to-noise ratio variations. Sedaghat et al. proposed the HOSS (histogram of oriented self-similarity) algorithm [18] based on self-similarity features, which ensures good rotational invariance but performs poorly in multi-modal images with high contrast and nonlinear radiometric distortions. The RRSS (rank-based ratio self-similarity) method [19] proposed by Xiong et al. effectively addresses differences between multi-modal images but incurs some loss in rotational invariance.

Some experts and researchers have tackled multi-modal image matching from the perspective of phase congruency models. However, it relies on a strategy involving circular feature calculations to overcome rotational differences, which results in lower technical efficiency. Xiang et al. [20] enhanced the PC model, keypoint extraction, and similarity measurement methods, constructing features for optical and SAR image matching. Fan et al. [21] designed a multi-scale PC descriptor known as multi-scale adaptive block phase congruency (MABPC). This descriptor utilizes multi-scale phase congruency features encoded with an adaptive block spatial structure. Yao et al. [22]. proposed absolute phase orientation histogram matching, designing absolute phase-oriented features to accommodate differences between multi-modal images and resist scale, displacement, and rotational differences. However, this method is limited to matching tasks with small rotational differences. Yang et al. [23] introduced local phase sharpness-oriented features to adapt to MMRSI matching and improve the applicability to rotational differences in multi-modal images but still lacks complete local rotational invariance.

Recently, various algorithms have been proposed to address multi-modal image differences by improving image scale space, such as the CoFSM algorithm [24], which reduces multi-modal image differences by improving the image scale space. Other approaches include the MS-HLMO algorithm [25], which is based on multiscale joint mean minimal gradient features, and multi-modal image matching through local normalized image filtering [26]. These methods have improved the matching performance of multi-modal remote sensing images, demonstrating strong rotational invariance. However, they still suffer from issues such as the low accuracy of corresponding points and poor matching robustness to varying degrees.

With the rapid development of artificial intelligence theory, deep learning techniques have been introduced into multi-modal image matching. Examples include convolutional neural network-based matching [27] and graph neural network-based matching (Super-Glue) [28]. These methods have demonstrated excellent performance in homogenous image matching but have shown limitations when applied to multi-modal images. In response to

this challenge, researchers have explored methods specifically designed for multi-modal image matching. These include the D2-Net network for multi-source image feature extraction and description [29], the LoFTR algorithm based on transformer networks [30], and its improved version, SE2-LoFTR, with rotational invariance [31]. M2DT-Net [32] is a method that combines learned features with Delaunay triangulation constraints, but it has certain limitations in handling radiometric invariance. Deep learning-based approaches offer speed and strong feature learning capabilities, especially in the context of multi-modal image matching, where they have shown significant potential. However, due to substantial variations in land features between multi-modal images and the difficulty in obtaining training samples, the generalization and applicability of such methods are currently limited.

In summary, MMIs exhibit significant differences in illumination, contrast, nonlinear geometric and radiometric distortions, making it challenging to extract reliable similarity features and often leading to matching failures. When registering multi-modal remote sensing images, traditional parameter optimization methods are prone to becoming stuck in local optima, resulting in lower registration accuracy that cannot meet the requirements of high-precision applications. We proposed a method that starts by extracting features through the design of a phase difference pyramid and then constructs feature descriptors using normalized filtering in logarithmic polar coordinates. This approach aims to improve the success rate and accuracy of multi-modal image matching. The method presented in this paper enhances the precision and robustness of feature matching and registration in multi-modal remote sensing images. It also satisfies the demands of high-precision tasks such as image fusion, image stitching, and change detection, holding significant theoretical and practical value.

## 3. Methods

The proposed PEDoG method consists of four steps: (1) the construction of the phase-indexed difference of gaussian pyramid, (2) improved SIFT-like feature point extraction, (3) the construction of normalized filtering in logarithmic polar coordinate descriptors, and (4) matching and outlier removal. Of these steps, (1) and (3) are the focal points of this paper. In step (1), we build the phase-indexed Gaussian difference pyramid by first calculating the maximum moment feature of the image using the phase congruency model. Next, we construct a Gaussian difference pyramid for the moment feature and then optimize the Gaussian phase pyramid using exponential equations, ultimately generating the phase-indexed Gaussian difference pyramid feature. The feature matching and outlier removal processes employ the nearest-neighbor ratio matching strategy [33] and the fast sample consensus algorithm [34] to eliminate outliers, respectively. Following in Figure 1, shows the overall flow chart of the proposed method.

### 3.1. Phase Exponent of Difference of Gaussian-Pyramid

To address the issues of nonlinear radiometric distortions and noise between multi-modal remote sensing images, a phase-indexed Gaussian difference pyramid feature detector is introduced. This detector utilizes the structure tensor-based phase congruency (PC) to construct a difference image pyramid that is robust to nonlinear radiometric distortions (NRD) and noise. The following sections provide a detailed explanation.

(1) Phase coherence model solving. The PC value represents the importance of features and is robust to NRD and noise [35]. Therefore, the PC structure map is very suitable for feature point detection in multi-modal remote sensing images. The PC structure map is calculated as follows. First, the image $I(x, y)$ is convolved with the odd-symmetric wavelet $L^{odd}(x, y, s, o)$ and even-symmetric wavelet $L^{even}(x, y, s, o)$ to obtain the response components $E_{so}(x, y)$ and $O_{so}(x, y)$.

$$\begin{cases} L(x, y, s, o) = L^{even}(x, y, s, o) + i \cdot L^{odd}(x, y, s, o) \\ [E_{so}(x, y), O_{so}(x, y)] = \left[ I(x, y) \otimes L^{even}(x, y, s, o), I(x, y) \otimes L^{odd}(x, y, s, o) \right] \end{cases} \quad (1)$$

where *s* and *o* denote the scale and orientation of Log–Gabor wavelets, respectively; $L^{even}(x,y,s,o)$ represents the even-symmetric filter of the Log–Gabor filter; $L^{odd}(x,y,s,o)$ represents the odd-symmetric filter; *i*-th represents the imaginary unit of the retest; $I(x,y)$ is the image; $E_{so}(x, y)$ represents the response result of the image on the real part filter; $O_{so}(x,y)$ represents the response result of the image on the imaginary part filter; and $\otimes$ represents the convolution operation.

Then, the amplitude component $A_{so}(x,y)$ at scale s and direction o is calculated. By combining all directions and scales and introducing a noise compensation term *T*, the value of $PC(x,y)$ is calculated as follows:

$$PC(x,y) = \frac{\sum_S \sum_O w_O(x,y) \lfloor A_{SO}(x,y)\Delta\Phi_{SO}(x,y) - T \rfloor}{\sum_S \sum_O A_{SO}(x,y) + \xi} \tag{2}$$

where $w_o$ is the weighting factor and $\xi$ is a small constant that prevents the denominator from being zero, and when its value is less than zero, $\lfloor . \rfloor$ means it is equal to zero, otherwise it is unchanged. $\Delta\Phi_{so}(x,y)$ is the phase deviation function and $A_{so}(x,y)\,\Delta\Phi_{so}(x,y)$ can be calculated from the response components $E_{so}(x,y)$ and $O_{so}(x,y)$. Also, it can be calculated using Equation (2), which can calculate the importance of the phase feature at each pixel and is then defined as the maximum moment.

(2) Construction of a Gaussian difference image pyramid. In order to extract scale-invariant feature points in multimodal remote sensing images, this paper constructs a differential image pyramid using the phase coherence structure map. First, the scale space $L(x,y,\sigma)$ is constructed from the PC structure map $PC(x, y)$ and the Gaussian function $G(u, v, \sigma)$ with the formula shown in (3) and the Gaussian function $G(u, v, \sigma)$ shown in (4):

$$L(x,y,\sigma) = M(x,y) \otimes G(u,v,\sigma) \tag{3}$$

$$G(u,v,\sigma) = e^{-\frac{u^2+v^2}{2\sigma^2}} \tag{4}$$

In the scale space, Gaussian-smoothed images are computed using a Gaussian function with continuous scale $\sigma_n = kn_{\sigma 0}$, where *k* is a constant multiplication factor between two neighboring scales. Then, the phase difference image pyramid is computed as shown in (5):

$$D(x,y,\sigma) = L(x,y,k\sigma) - L(x,y,\sigma) \tag{5}$$

In Equation (5), denotes the phase Gaussian difference pyramid result. However, in order to further enhance the structural feature information of the image and reduce the contrast difference of the image, this paper also introduces the exponential function [36] to optimize the phase Gaussian difference pyramid features, the expression of which is shown in (6):

$$PE(x,y,\sigma) = -[\exp(D(x,y,k\sigma)) - \exp(D(x,y,\sigma))] \tag{6}$$

In Equation (6), $PE(x,y,\sigma)$ denotes the phase-exponential Gaussian difference pyramid result. In order to better demonstrate the results of the phase-exponential Gaussian difference pyramid, a set of infrared–optical images are selected to be significant in this paper, and the results are shown in Figure 2.

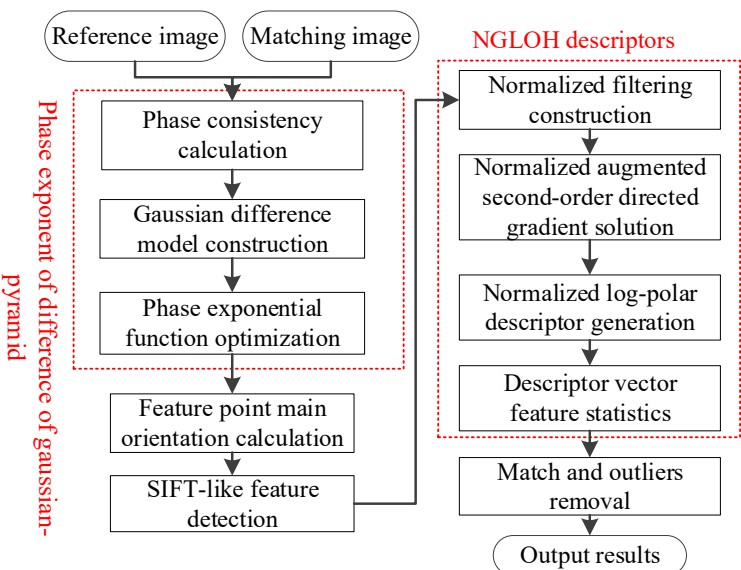

**Figure 1.** Overall flow chart.

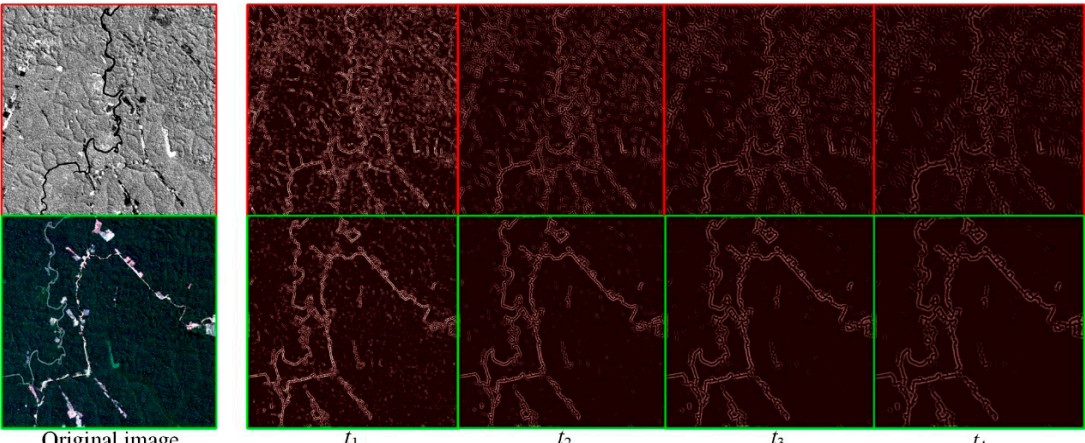

**Figure 2.** Phase-exponential Gaussian differential scale space.

### 3.2. Improved Feature Detection

After constructing the difference image pyramid, the detection of extremal points within its multi-scale octave range is performed. Typically, it involves comparing a pixel with 26 points in its neighborhood, as seen in algorithms like SIFT [5]. However, the texture of the PC structural map is quite sparse, and the Gaussian function further blurs it, resulting in the detection of only a few feature points. Therefore, it is possible to directly use an 8-neighborhood detection method to ensure an adequate number of feature points.

First, in each layer of the DoG image pyramid, candidate feature points with locally very large or very small values within their 8 neighborhoods are detected. In addition, the feature point extraction threshold ($C_t$) is set as its mathematical expression is shown in (7):

$$\begin{cases} PE(x,y,\sigma_n) > PE(x',y',\sigma_n) \\ PE(x,y,\sigma_n) < PE(x',y',\sigma_n) \\ \quad PE(x,y,\sigma_n) > C_t \end{cases} \tag{7}$$

where $(x',y')$ denotes the coordinates of the 8 neighborhood pixel points of $(x,y)$. Then, it is verified whether the phase index of the candidate feature point in the Gaussian difference detector in the scale direction is greater than the feature point extraction threshold value.

Based on the above three steps, the phase-exponential Gaussian differential feature detector can extract scale-invariant feature points. As can be seen in Figure 3, the feature points are basically distributed over the image structure and a large number of corresponding feature points are detected in multimodal remote sensing image pairs.

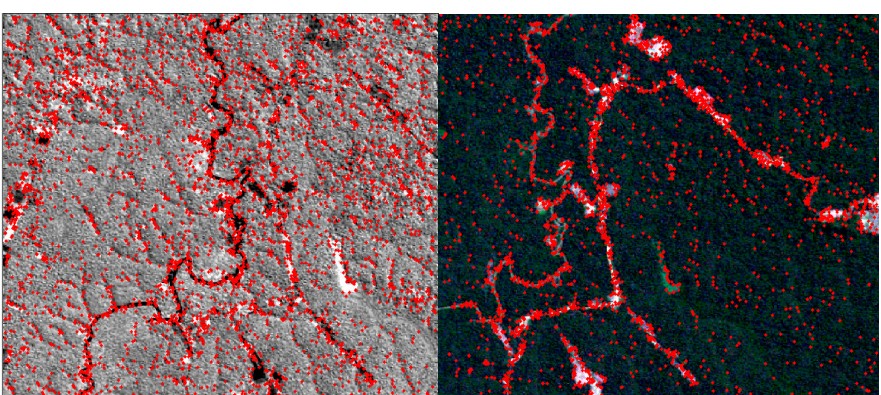

**Figure 3.** Schematic distribution of feature points.

### 3.3. Improved GLOH-like Feature Descriptors

In this paper, we design improved log-polar descriptors for normalized filtering. Firstly, normalized filtering is introduced to weaken the modal differences of the image, and then the second-order oriented gradient features after normalized enhancement are solved. Then, the joint log-polar coordinate descriptor framework calculates the descriptor vector features to complete the descriptor construction. The flowchart of an improved log-polar descriptor is shown in Figure 4.

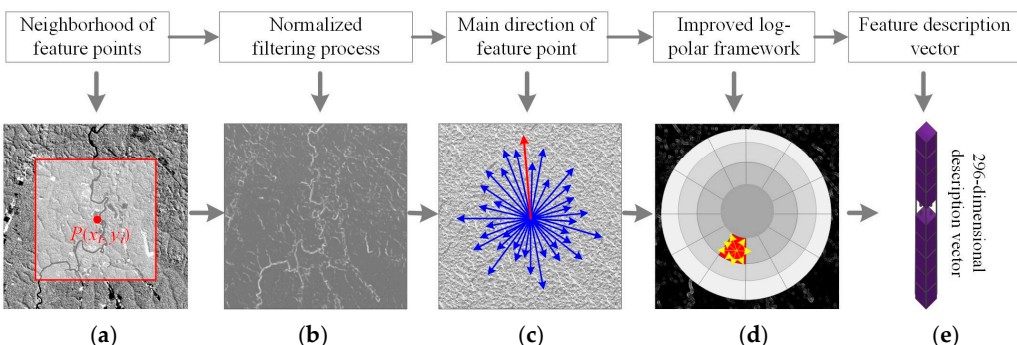

**Figure 4.** Normalized log-polar coordinate descriptor flowchart. (**a**) is the neighborhood window, (**b**) is the result of the normalization process, (**c**) is the feature principal direction, (**d**) is the description subframe, and (**e**) is the feature vector value.

#### 3.3.1. Normalized Filtering of Image-Oriented Gradient Features

Image-oriented gradient features are a crucial step in feature description. However, traditional gradient features of images are sensitive to the nonlinear radiometric distortions in MMIs, making them less robust and less favorable for robustly representing feature points. Normalized filtering is a filtering method proposed by Li et al., 2022 [26], which effectively enhances the similarity information in MMIs. Building on this, we introduce this method into the computation of oriented gradient features for MMIs, thereby reducing modality feature differences and better expressing the common information in the images. The mathematical expression for normalized filtering is shown below:

$$P_N(x,y) = P(x,y) - \frac{1}{|W(x,y,s)|} \sum_{W(x,y,s)} P(x,y) \tag{8}$$

where $P_N(x,y)$ denotes the normalized image; $P(x,y)$ denotes the original image; $(x,y)$ denotes the coordinates of the image; $W(x,y,s)$ is a local window centered on $(x,y)$ with a size of $(2 \times s + 1) \times (2 \times s + 1)$; and $|\cdot|$ denotes the taking of an absolute value. Essentially, the local normalization filter is equivalent to the original image minus its average filtering result. Therefore, this filter retains only the detailed components, i.e., the normalized image contains most of the structural information of the image, which is very important for multimodal image matching. This is the important theoretical reason why we introduce this method.

Image-oriented gradient computation. The normalized filtered image is passed through the Soble operator [37] to eliminate the nonlinear brightness difference of the image. Therefore, in order to further highlight the feature information, the second- and third-order gradient computation of the image is carried out based on the joint Sobel template with the following equation.

$$
\begin{cases}
PS_\sigma^2 = \sqrt{\left(P_N(x,y)_{x,\sigma}^1 \cdot \Lambda_x\right)^2 + \left(P_N(x,y)_{y,\sigma}^1 \cdot \Lambda_y\right)^2} \\
Angle_\sigma^2 = \arctan\left(\frac{P_N(x,y)_{y,\sigma}^1 \cdot \Lambda_y}{P_N(x,y)_{x,\sigma}^1 \cdot \Lambda_x}\right)
\end{cases}
\tag{9}
$$

$$
\begin{cases}
PS_\sigma^3 = \sqrt{\left(PS_{x,\sigma}^2 \cdot \Lambda_x\right)^2 + \left(PS_{y,\sigma}^2 \cdot \Lambda_y\right)^2} \\
Angle_\sigma^3 = \arctan\left(\frac{PS_{y,\sigma}^2 \cdot \Lambda_y}{PS_{x,\sigma}^2 \cdot \Lambda_x}\right)
\end{cases}
\tag{10}
$$

$$
\Lambda_x = \begin{bmatrix} -1 & 0 & 1 \\ -2 & 0 & 2 \\ -1 & 0 & 1 \end{bmatrix} \qquad \Lambda_y = \begin{bmatrix} -1 & 2 & 1 \\ 0 & 0 & 0 \\ -1 & -2 & -1 \end{bmatrix}
\tag{11}
$$

In Equations (9)–(11), $PS_\sigma^2$ denotes the new second-order gradient amplitude; $Angle_\sigma^2$ denotes the new second-order gradient direction; $PS_\sigma^3$ denotes the new third-order gradient amplitude; $Angle_\sigma^3$ denotes the new third-order gradient direction; $\Lambda_x$ and $\Lambda_y$ are the gradient order amplitude function, where $\Lambda_x$ denotes the X-direction Sobel operator template; $\Lambda_y$ denotes the Y-direction Sobel operator template; and $\sigma$ denotes the image scale.

### 3.3.2. Log-Polar Descriptive Feature Solving

After extracting key points, describing them is a critical subsequent step for successful matching. The log-polar histogram framework is a classical framework that has been successfully applied in multi-modal image matching. The log-polar descriptor approach using the gradient location-orientation histogram (GLOH) algorithm is notably advantageous and stable [38]. However, the log-polar descriptor method is not the only one available, and it heavily depends on the partitioning of the polar grid. Different partitioning methods can generate different descriptors. Therefore, it exhibits better versatility when constructing descriptors.

Therefore, considering the stability and robustness of the descriptor, this paper, starting from the zero-degree direction on the right end of the polar grid, divides the circular neighborhood into 12 equal parts, creating a new polar coordinate grid consisting of $(12 \times 3 + 1)$ sub-region grids. Each sub-region in the polar coordinate grid has approximately equal area. Within each grid, the horizontal direction represents the polar angle of the pixels within the circular neighborhood. After calculating the orientation histogram for each feature point, the direction is divided into 8 dimensions at 45-degree intervals, covering a range from 0 to 360 degrees. As a result, each sub-region grid has adjacent points with 8-dimensional gradient location-orientated histograms. Finally, multiplying the number of sub-region grids in the log-polar coordinates (38) by the dimensions of the orientation histogram (8 dimensions) generates a novel 296-dimensional log-polar descriptor.

## 4. Results

To evaluate the performance of PEDoG, it was compared to five state-of-the-art algorithms (SIFT, PSO-SIFT, RIFT2 [39], OSS (oriented self-similarity) [40], and HOWP (histogram of the orientation of the weighted phase descriptor) [3]). To ensure a fair comparison, the code provided by the authors and their recommended parameter settings were used. The paper employed the fast sample consensus algorithm to eliminate mismatches and obtain corresponding points. PEDoG was implemented in MatlabR2020a, and the experimental platform used an AMD Ryzen 9 5900HX with Radeon Graphics processor clocked at 3.30 GHz, 64 GB of RAM, and the Windows 11 X64 operating system.

### 4.1. Experimental Datasets

In this study, three commonly used multi-modal data types were selected as data sources, including multi-temporal images, infrared and optical images, and electronic navigation maps with optical images, totaling 60 image pairs, with each modality consisting of 20 image pairs. The image sizes ranged from 381 pixels to 750 pixels. One pair of images from each data type is shown in Figure 5. This dataset encompasses diverse scenarios for multi-modal image matching applications, with ground resolutions ranging from 1 m to 300 m, making it representative. Additionally, significant nonlinear radiometric differences and geometric transformation differences exist between the image pairs.

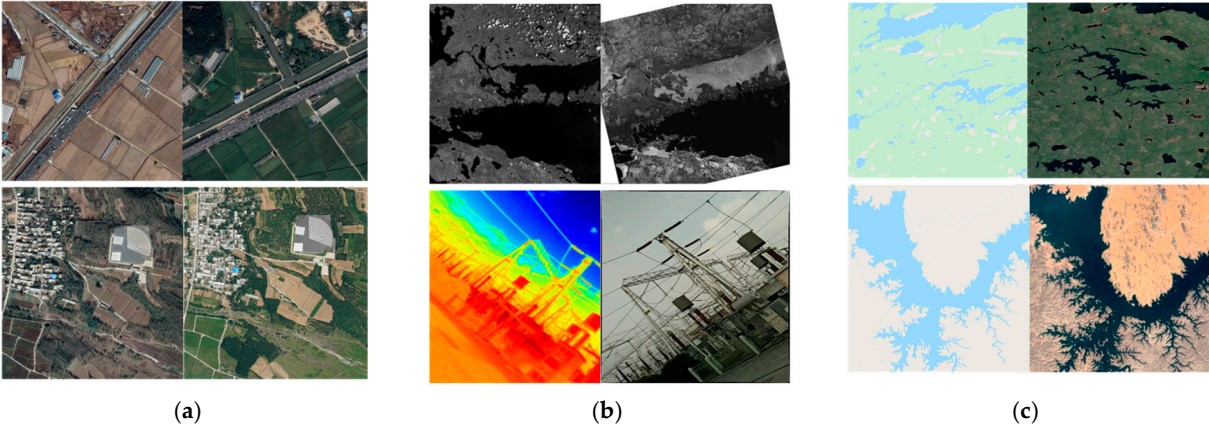

(**a**)　　　　　　　　　　　　(**b**)　　　　　　　　　　　　(**c**)

**Figure 5.** Partial multi-modal image data. (**a**) optical–optical; (**b**) infrared–optical; (**c**) map–optical.

### 4.2. Evaluation of Indicators

To comprehensively evaluate the performance of the proposed PEDoG method, this paper provides both qualitative and quantitative comparative results. The quantitative evaluation employs three metrics to assess the matching performance of the methods: success rate (SR), NCM, and RMSE. SR measures the ratio of successfully matched image pairs to the total number of image pairs. This metric reflects the robustness of the matching method for specific types of multi-modal image pairs. Image pairs with an NCM of less than 20 and matching errors greater than seven pixels are considered matching failures. The number of matches within seven pixels are counted as the correct matches. RMSE is a commonly used metric to assess the performance of matching algorithms. RMSE reflects the precision of correct matches. A smaller RMSE value indicates higher accuracy. The mathematical expression for RMSE is given in Equation (12):

$$RMSE = \sqrt{\frac{1}{N}\left(\sum_{i=1}^{N}\left[(x_i' - x_i'')^2 + (y_i' - y_i'')^2\right]\right)} \tag{12}$$

where $N$ is the number of homonymous points, $(x_i'', y_i'')$ is the coordinate of the *i*-th truth point $(x_i', y_i')$ after the corresponding matching transformation.

*4.3. Results*

Table 1 displays the average results for the four metrics. In Figure 6a, "Image Type 1" represents multi-temporal optical–optical images, "Image Type 2" represents infrared–optical images, and "Image Type 3" represents map–optical images. In Figure 6b, the symbol "+∞" indicates matching failures or RMSE ≥ 7 pixels. The units for SR are percentage (%), NCM is in point numbers, and RMSE is in pixels.

**Table 1.** Findings of the six methodologies for the three evaluation indicators.

|      | SIFT  | PSO-SIFT | OSS   | HOWP   | RIFT2  | PEDoG  |
|------|-------|----------|-------|--------|--------|--------|
| SR   | 58.3% | 88.3%    | 91.7% | 66.7%  | 96.7%  | 100%   |
| NCM  | 79.53 | 139.95   | 175.6 | 236.92 | 209.38 | 259.43 |
| RMSE | 4.08  | 2.36     | 2.08  | 3.63   | 2.07   | 1.69   |

Figure 6 presents a comparison of the matching results between the PEDoG algorithm and five other algorithms using two metrics, NCM and RMSE. The results of the SIFT method are represented by a black dashed line. Its SR is the lowest, standing at only 58.3%, with an average NCM of 79.53, the lowest among the six algorithms. The RMSE is 4.08 pixels, indicating the lowest matching accuracy. The SIFT algorithm achieves feature description by computing the gradient features of images, but it is sensitive to multi-modal remote sensing images with nonlinear radiometric distortions and contrast differences, leading to matching failures. Therefore, the applicability of the SIFT algorithm in multi-modal remote sensing image matching is relatively poor.

The results of the PSO-SIFT algorithm are shown as a blue dashed line. It slightly outperforms the SIFT algorithm in terms of matching performance with an SR of 88.3% and an average NCM of 139.95. It has good MRSI scaling and rotation invariance, as evidenced by an average RMSE of 2.36 pixels.

The results of the OSS algorithm are indicated by a green dashed line. The algorithm utilizes offset mean filtering to quickly compute self-similar features. SR results are better at 91.7%, which is in the middle of the range. The average NCM is 175.6, which is lower than the HOWP, RIFT2, and PEDoG algorithms. This may be due to the low discriminative power of the LSS descriptors, which cannot maintain a robust matching performance in multimodal matching situations such as e-navigation maps and optical images. The RMSE of this algorithm is lower among the six algorithms with an average RMSE of 2.08 pixels.

The results of the HOWP algorithm are indicated by the purple dashed line. The successful matching rate SR of this algorithm is 66.7%, and the reason for the low successful matching rate is that the algorithm does not support multimodal remote sensing image matching with too large a rotation angle and lacks robust rotational invariance. The average NCM is 236.92 and the average RMSE is 3.63 pixels.

The results of the RIFT2 algorithm are indicated by the brown dashed line. The algorithm uses image frequency domain features to complete the matching and greatly optimizes the matching performance by maximizing the indexed map descriptors. The SR is increased to 96.7% and the average NCM is 209.38. the average RMSE is 2.07 pixels. the algorithm is altered from the RIFT algorithm to support rotational invariance but does not support scale differences, so the RMSE results are volatile.

The results of the PEDoG algorithm are represented by a solid red line, and it produced the most robust matching outcomes. The PEDoG algorithm successfully matched all 60 pairs of multi-modal remote sensing images, achieving a 100% SR. The average NCM reached the highest value at 259.43, and the average RMSE was 1.69 pixels. The PEDoG algorithm is capable of handling geometric variations, nonlinear radiometric differences, illumination discrepancies, and contrast differences in MRSI, while also possessing scale and rotation invariance. In summary, the PEDoG algorithm generates more robust matching results in multi-modal remote sensing image matching tasks and exhibits superior overall performance compared to the other five algorithms.

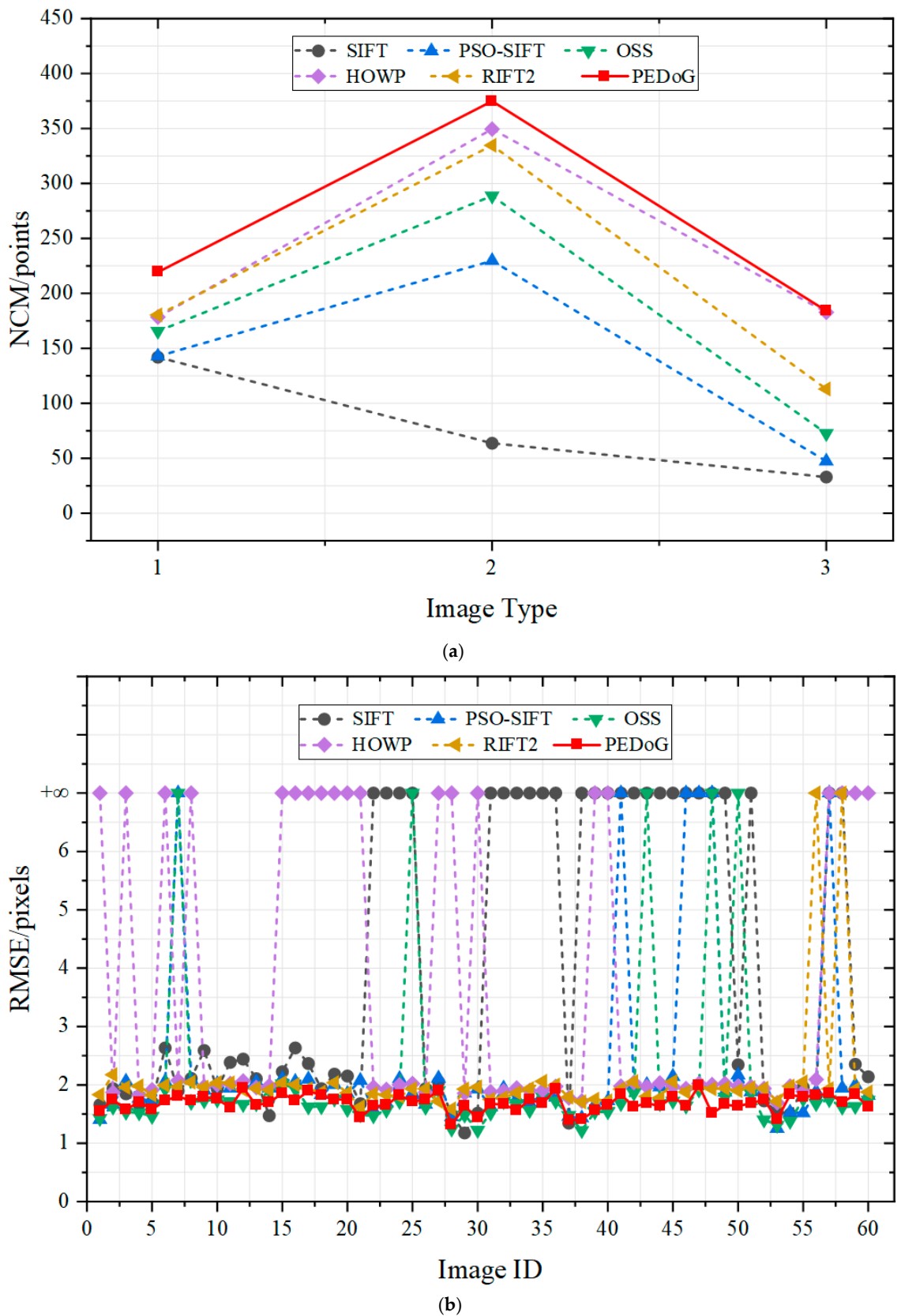

**Figure 6.** Test results of the six methods. (**a**) NCM results for six methods, (**b**) RMSE results for six methods.

(1)    Qualitative results

In order to better demonstrate the performance of the proposed PEDoG algorithm, this paper also qualitatively evaluates it against five comparison algorithms, and the results are shown in Figure 7.

Figure 7 shows the matching results of the six matching algorithms on the three types of data; the green line in the figure represents a correct match for the homonymous points, and the red line indicates a failed match. The optical–optical data show two sets of optical image pairs of different years; the four images shown in the infrared–optical data, where the image on the top left is an infrared image, the image on the bottom left is a thermal infrared image, and the matching images on their right are all optical images; in the map–optical data, the two on the left are electronic navigation maps, and the two on the right are the corresponding optical images. As shown in Figure 7, the proposed PEDoG algorithm can obtain the most abundant matching keypoints among the three modalities and exhibits good scale and rotation invariance. In contrast, the SIFT algorithm and PSO-SIFT algorithm perform well with multi-temporal images and electronic navigation maps but have only moderate performance with infrared images. The RIFT2 algorithm is an improved version of RIFT, demonstrating good rotation invariance but not ideal scale adaptation. The OSS algorithm employs local self-similarity descriptors and performs well with both multi-temporal images and infrared images but yields fewer matching keypoints. Lastly, the HOWP algorithm uses weighted phase-oriented features for description and performs well, although it is sensitive to significant rotational differences.

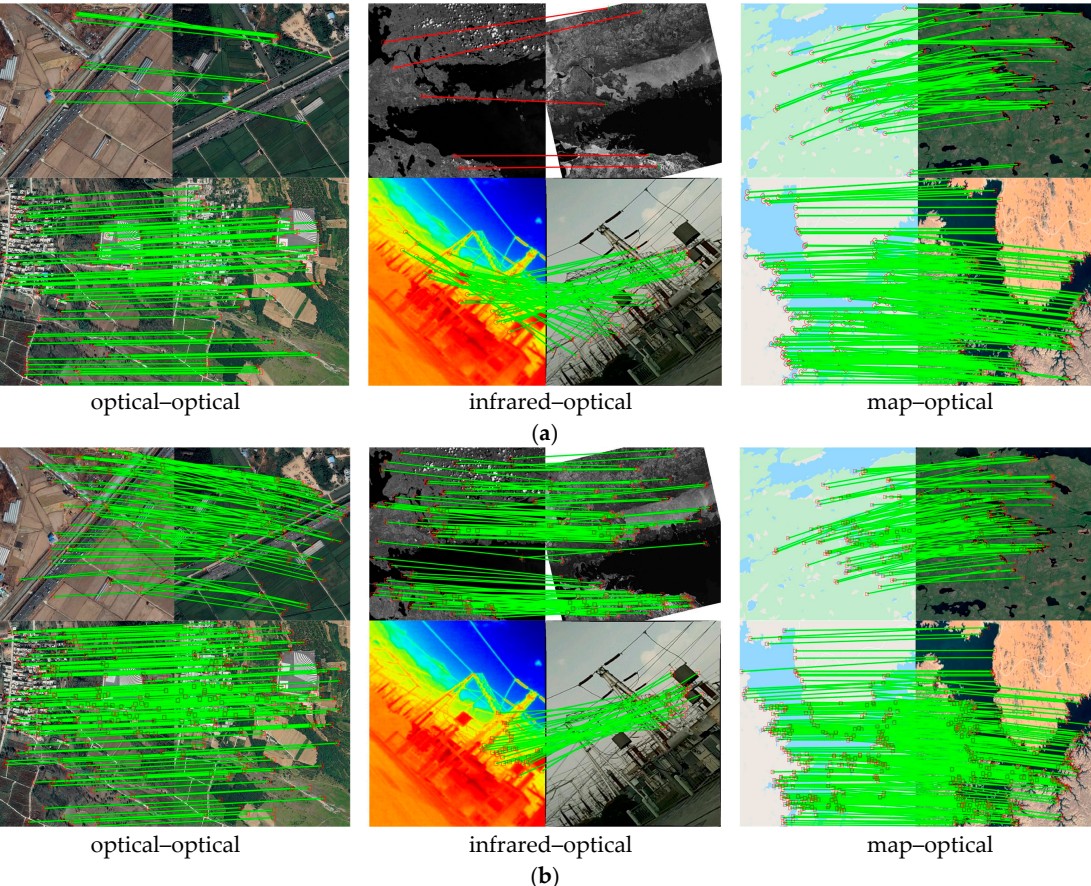

optical–optical                    infrared–optical                    map–optical

(**a**)

optical–optical                    infrared–optical                    map–optical

(**b**)

**Figure 7.** *Cont.*

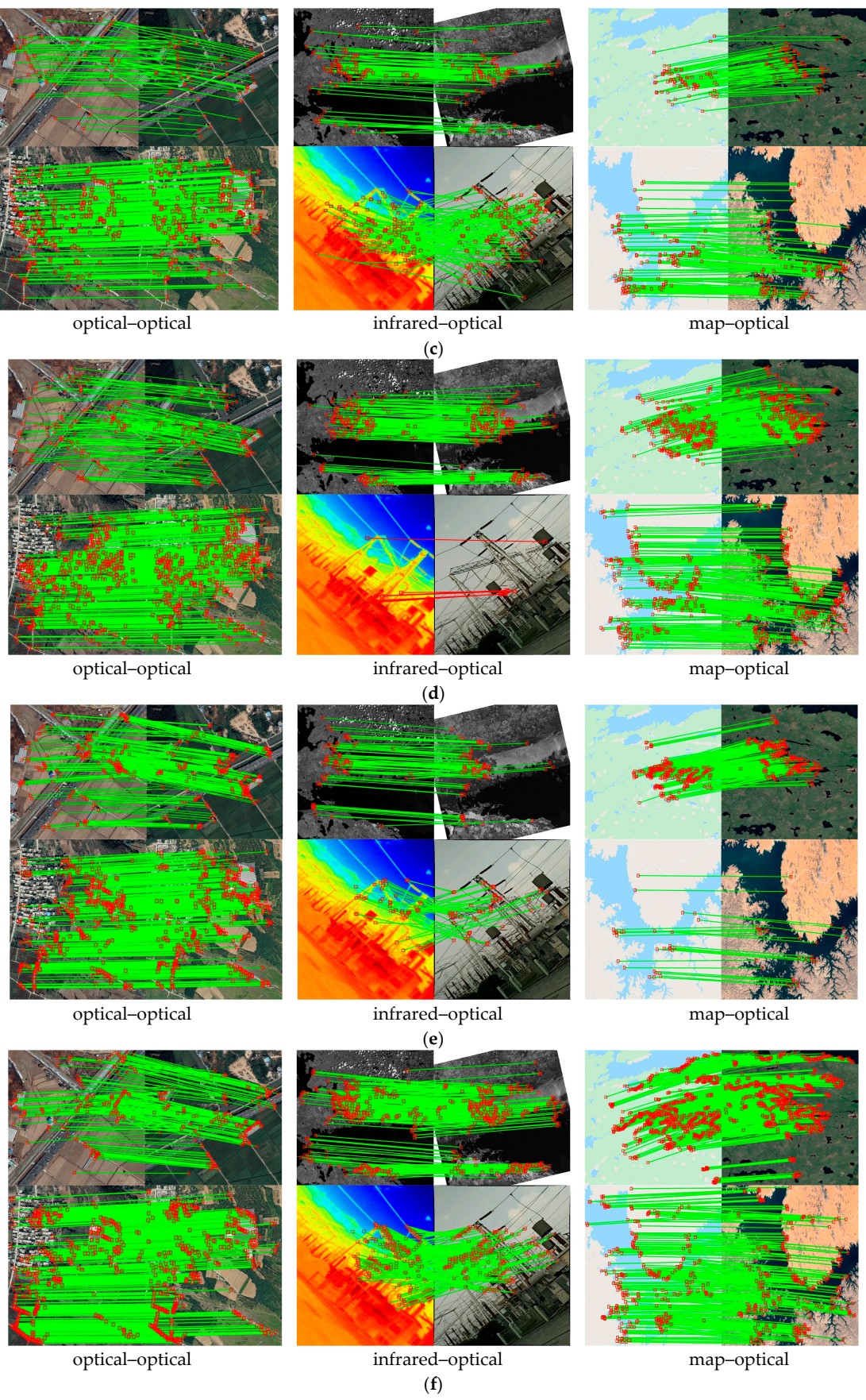

**Figure 7.** Qualitative matching results for six matching algorithms. (**a**) SIFT; (**b**) PSO-SIFT; (**c**) OSS; (**d**) HOWP; (**e**) RIFT2; (**f**) PEDoG.

(2)   Results of the registration

To further demonstrate the performance of the PEDoG algorithm, a checkerboard visualization method was employed to showcase the registration results, as depicted in Figure 8. In the displayed image checkerboards, it is evident that the overlapping regions between images are fully aligned, effectively avoiding artifacts and misalignment issues. This further underscores the algorithm's high registration accuracy. The exceptional matching performance of the PEDoG algorithm can be attributed to two main factors. Firstly, the PEDoG algorithm extracts keypoints using the phase-difference Gaussian pyramid, effectively leveraging the similarity features in multi-modal images, resulting in keypoints with higher repeatability and saliency. Secondly, the designed normalized log-polar coordinate descriptors, by introducing normalized filtering, effectively reduce the sensitivity of image gradient features to nonlinear radiometric distortions, thus achieving the superior representation of multi-modal images and enhancing registration accuracy.

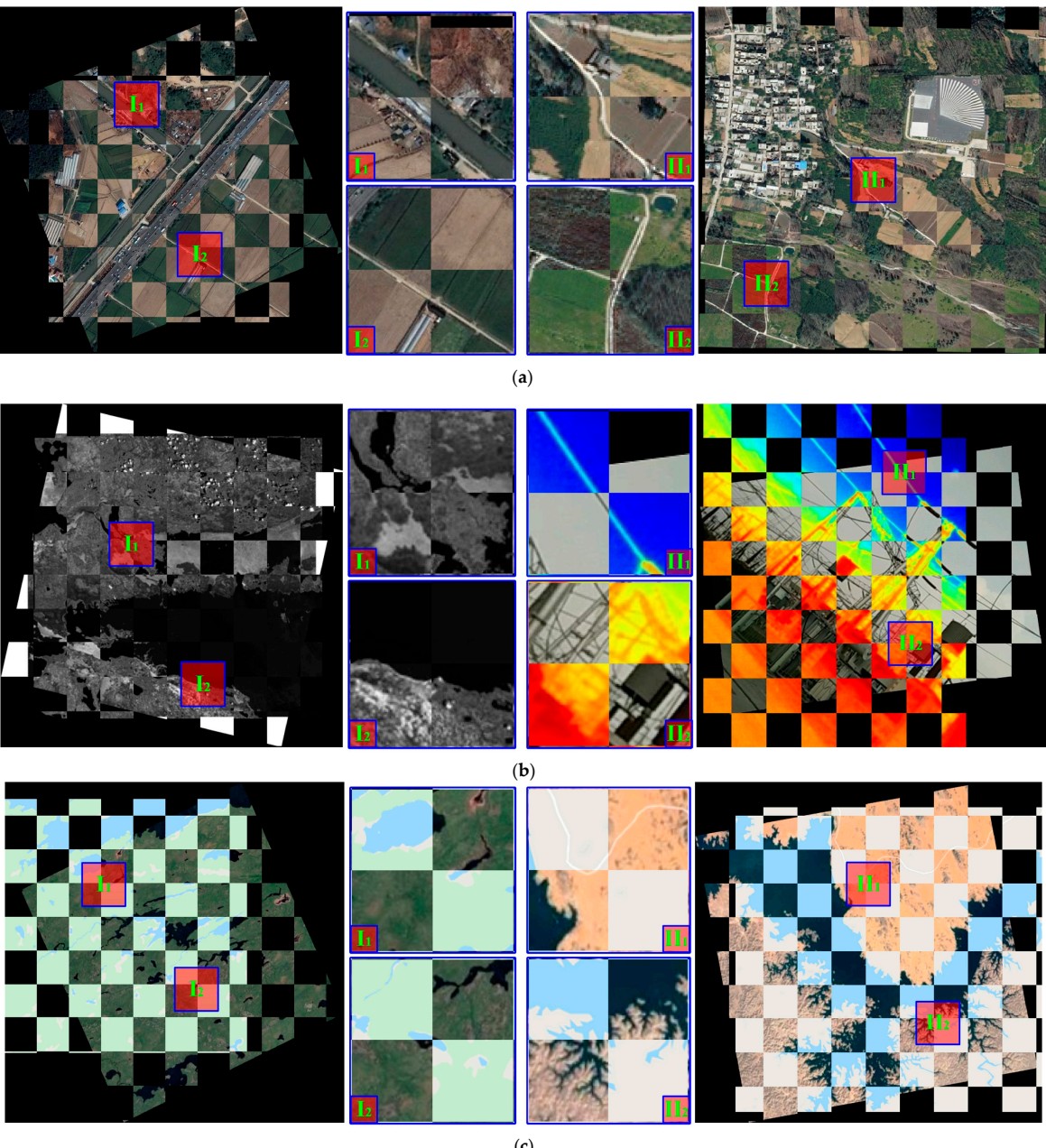

**Figure 8.** Registration results for selected images. (**a**) multi-temporal–optical; (**b**) infrared–optical; (**c**) map–optical.

## 5. Analysis and Discussion

### 5.1. Parameter Setting

To comprehensively evaluate the performance of the PEDoG algorithm, an analysis of its two core parameters was conducted: the number of layers in the differences of the Gaussian (DOG) pyramid, denoted as $D_l$, and the feature point extraction threshold $C_t$. RMSE and NCM were employed as evaluation metrics to assess the impact of these parameters. The specific parameter settings are presented in Table 2. Through parameter tuning and experimental evaluations, we identified the optimal parameter configuration that achieves the best matching performance. Such parameter selection not only leads to ideal results in terms of RMSE and NCM but also ensures that the matching task can be completed within an acceptable time frame. RMSE is measured in pixels, while NCM is measured in point counts.

**Table 2.** Recommended parameter settings for the PEDoG algorithm.

| Parameter | Variable Values | Fixed Parameters |
|---|---|---|
| $D_l$ | $D_l$ = [2, 3, 4, 5, 6, 7, 8] | $C_t$ = 0.3 |
| $C_t$ | $C_t$ = [0.1, 0.2, 0.3, 0.4, 0.5, 0.6, 0.7, 0.8, 0.9] | $D_l$ = 4 |

In this paper, 60 sets of MMI data are used to test the performance of PEDoG under different parameter settings. Figure 9a demonstrates the matching results under different DOG pyramid layers $D_l$. As $D_l$ increases from 2 to 5, the NCM results show a steady upward trend to 268. However, when Dl exceeds 5, the NCM starts to decrease slightly until $D_l$ reaches 8, at which point the NCM stays around ~250. For RMSE, the lowest value was reached at $D_l$ = 4, which was 1.69 pixels. Considering NCM and RMSE together, the highest accuracy and rich matching of homonymous points can be obtained when $D_l$ is set to 4, so it is recommended to set $D_l$ to 4.

Figure 9b demonstrates the results for different coarseness rejection threshold $C_t$ parameter settings. From the figure, it can be observed that the number of NCMs is decreasing as $C_t$ increases, and considering the RMSE factor, the optimal parameter setting should be 0.3. The RMSE is at a low level while more matches are obtained. Therefore, in this paper, the number of pyramid layers $D_l$ = 4 and the coarseness rejection threshold $C_t$ = 0.3 are chosen as the parameter settings that match the richness of corresponding points with optimal accuracy.

### 5.2. Analysis of Rotational-Invariance

To verify the performance of the PEDoG algorithm in terms of rotation invariance, the same set of images was used for matching tests. Initially, the reference image was rotated in both clockwise and counterclockwise directions at 30-degree intervals, resulting in a total of 12 simulated images across six different orientations. Subsequently, these 12 sets of simulated images were used for matching tests, and the matching result images were displayed. As shown in Figure 10 ("-" represents counterclockwise rotation), in a range of [−180°, 180°] for the rotation difference setting, the PEDoG algorithm consistently achieved successful matches and obtained abundant NCM. This is attributed to the fact that the proposed PEDoG algorithm first utilizes the exponential phase difference pyramid to extract feature points. This approach ensures the extraction of significant feature points in multi-modal images while reducing the sensitivity of gradient features to nonlinear radiometric distortions, thus ensuring the accurate calculation of feature point primary directions. Furthermore, the design of normalized log-polar descriptors further mitigates the modality differences between multi-modal images, enhancing the feature description capability between such images and ensuring robustness in matching.

In summary, the PEDoG algorithm possesses rotational invariance. The proposed PEDoG algorithm exhibits the most robust MMI matching performance in terms of handling rotational invariance as well as matching accuracy compared to traditional methods.

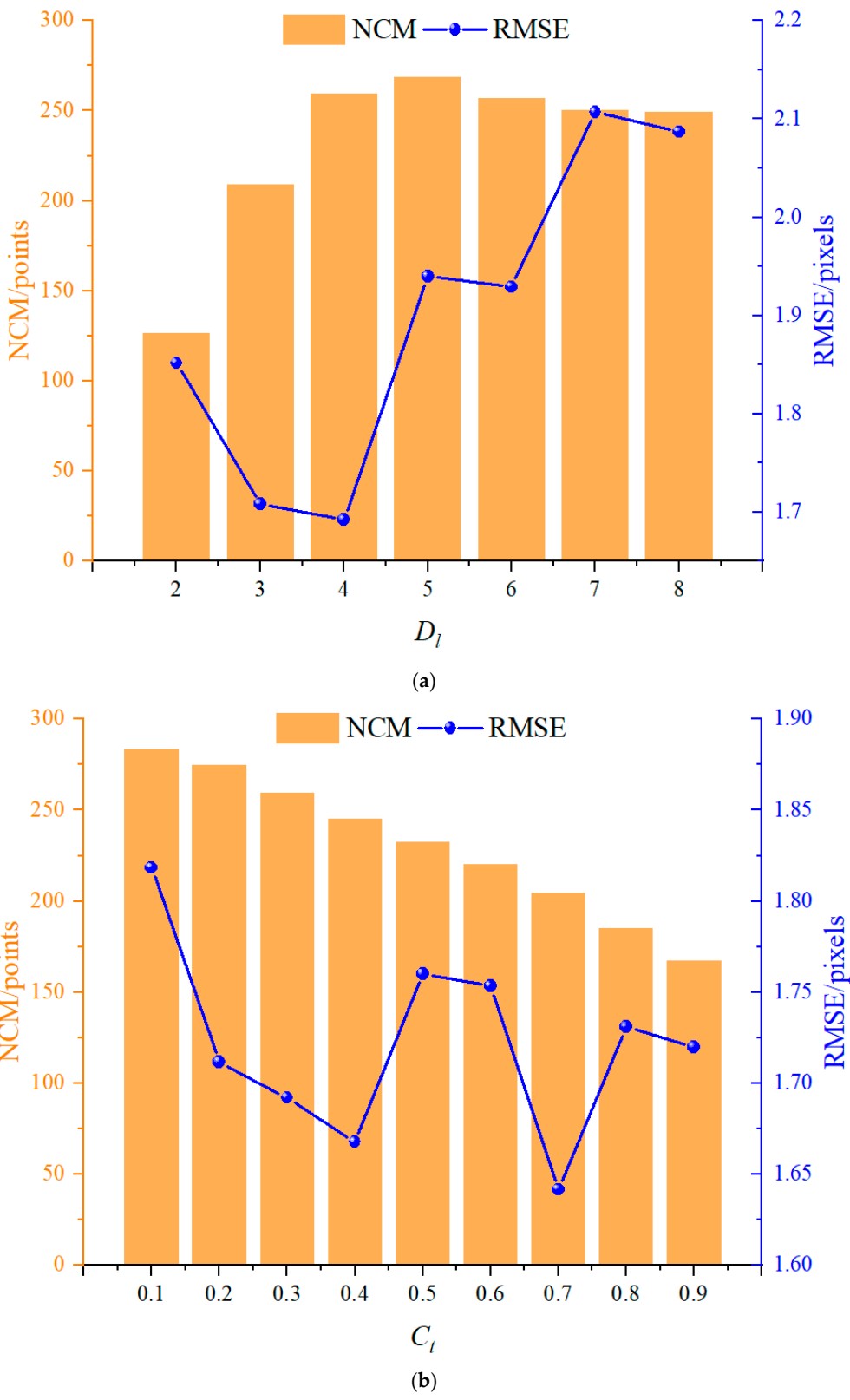

**Figure 9.** Parameter analysis results. (**a**) Results for parameter $D_l$; (**b**) Results for parameter $C_t$.

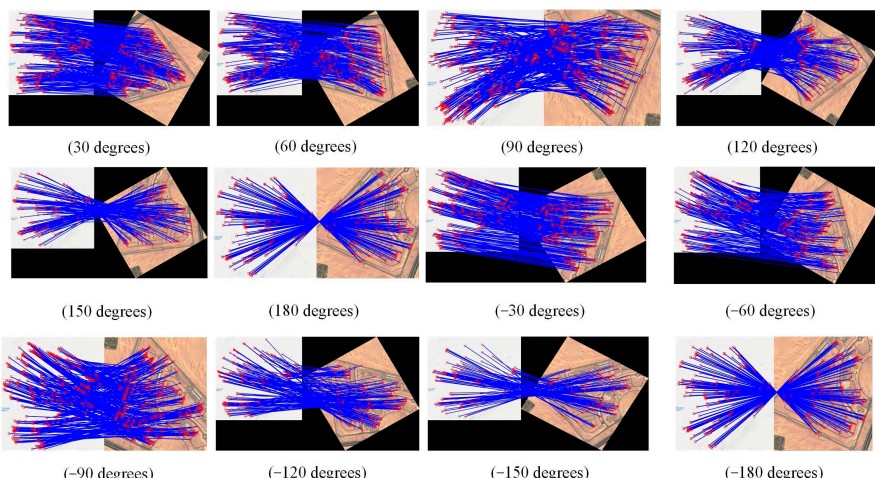

**Figure 10.** Rotation invariant matching results.

*5.3. Analysis of Scale-Invariance*

To verify the performance of the PEDoG method in terms of scale invariance, we used a set of map–optical image pairs for matching tests. First, we processed the reference images at 0.2× image scale intervals to generate six sets of simulated images, ranging from 0.6× to 1.8×. The matching results are shown in Figure 11. The observation of the results shows that the number of NCMs obtained decreases as the scale difference increases. Among them, the 1.8-fold scale yields the least number of matching points, but it is sufficient for the alignment operation. Therefore, the PEDoG method is effective in achieving scale invariance for MMI matching.

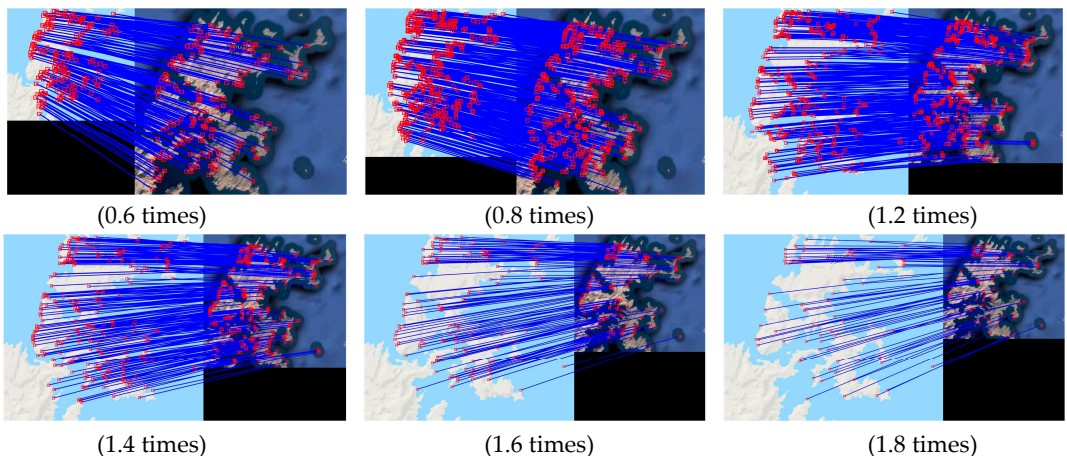

**Figure 11.** Scale invariant matching results.

**6. Conclusions**

In multi-modal images, the differences in imaging mechanisms often lead to significant variations in signal-to-noise ratios (SNR) among different modalities. This, in turn, poses a challenge for matching algorithms to maintain both matching geometric invariance and accuracy. Therefore, achieving high-precision matching correspondences while mitigating SNR interference in multi-modal remote sensing image matching is a challenge. To address this challenge, this paper proposes a lightweight multi-modal image registration algorithm that incorporates phase-indexed difference pyramids and normalized filtering. It aims to preserve geometric invariance while achieving high-precision matching correspondences. The proposed PEDoG matching method consists of three main parts.

The introduction of the phase consistency model into the difference Gaussian pyramid, resulting in the creation of new phase indices. The selection of four types of multi-modal



remote sensing images (multi-temporal images, infrared–optical images, electronic navigation, and map–optical images) as experimental datasets. Comparison with state-of-the-art matching methods (SIFT, PSO-SIFT, OSS, RIFT2, and HOWP). The results indicate that the PEDoG algorithm achieves significant improvements, with an SR increase of up to 41.7% compared to the SIFT algorithm. Furthermore, its NCM is 3.3 times that of the SIFT method, 1.9 times that of the PSO-SIFT method, 1.5 times that of the OSS method, 1.1 times that of the HOWP method, and 1.2 times that of the RIFT2 method. Additionally, the RMSE achieved by PEDoG is the lowest among all matching methods, with a value of 1.69 pixels.

Finally, the registration results are demonstrated using a checkerboard mosaic pattern, showing consistent alignment of image features without artifacts or misalignment. Thus, the PEDoG algorithm proposed in this paper achieves high-precision registration while preserving geometric invariance.

**Author Contributions:** Conceptualization, X.Y. and Y.Y. (Yongxiang Yao); methodology, X.Y.; software, X.Y.; validation, X.Y., Y.Y. (Yongxiang Yao) and Y.C.; formal analysis, Y.C.; investigation, X.Y.; resources, X.Y.; data curation, Y.C.; writing—original draft preparation, X.Y.; writing—review and editing, X.Y.; visualization, Y.Y. (Yongxiang Yao); supervision, Y.Y. (Yijun Yangand); project administration, Y.Y. (Yijun Yangand); funding acquisition, Y.Y. (Yijun Yangand). All authors have read and agreed to the published version of the manuscript.

**Funding:** This research was funded by the National Natural Science Foundation of China under grant 62102268; the Stable Supporting Program for Universities of Shenzhen under grant 202208012102547001; the Research Foundation of Shenzhen Polytechnic University under grants 6022312044K, 6023310030K, 6021310008K, and 6019310010K; and the Post-doctoral Later-stage Foundation Project of Shenzhen Polytechnic University under grants 6020271005K and 6021271004K.

**Data Availability Statement:** Data are contained within the article.

**Conflicts of Interest:** The authors declare no conflict of interest. The funders had no role in the design of the study; in the collection, analyses, or interpretation of data; in the writing of the manuscript; or in the decision to publish the results.

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
