# Peer review of "Multi-Modal Image Registration Based on Phase Exponent Differences of the Gaussian Pyramid"

_remotesensing, doi:10.3390/rs15245764_

Round 1

Reviewer 1 Report

Comments and Suggestions for Authors

The paper presents a new method for registration of images collected with different sensor types to map the points between the different images. A new method using the Gaussian image pyramid using phase exponents of differences from the Gaussian image pyramid was used to find the corresponding points.  Results are presented showing that the proposed method does give better results than the previously proposed methods for a set of images.

Overall, I think that this is a high quality paper but I do have a few issues with the presentation that I think should be corrected prior to the final submission.

1. On page 5, the functions and elements are used in the equations without any proper definitions.  Many of the times can be found through context but it is standard to define all items in equations in academic papers.  For example, the item p_{s,o} (x,y) is not defined.  The functions Lodd(x,y,s,o) and Leven(x,y,s,o) are referred to in the text but not used in the following equations.  The operator \ocross needs to defined as well.  It seems to refer to convolution but this should be clearly defined somewhere.

2. Equation 3 uses a PC structure map, M(x,y) and Gaussian function G(u,v,\sigma) but these are never explicitly defined in the paper. 

3. On page 6, it is said that a pixel is compared with 27 points in its neighbourhood.  What are these 27 points?  How are they decided?

4. On page 7 and page 8, Gradient order amplitude functions are used but these have not been explicitly defined.

5. On pages 11-13, some explanation should be given as to what Figure 7 is actually showing.  What do the green lines represent? What are the 4 images in each sub-figure? It is not to hard to figure out from context but the reader should be given some guidance.

6. On page 16, the outlier rejection threshold C_t is shown to have a major effect on performance but this value is not seen in any of the prior equations or descriptions of the algorithm.  What exactly is this value and why does it effect the performance?  

7. More explanation is needed about the selection of the images used to test the new method.  What evidence is there that the test images will have the characteristics of actual images that practitioners are likely to encounter in field applications of the algorithm?  Are these images from standard test sets for this type of algorithm.  Some assurance should be given to the reader that the image selection was based on characteristics from real sensor image collection and not a selection of images that would favour the proposed algorithm. I believe that the authors actually did pick a fair data set but the paper should give some explanation to back this assertion.

Overall, I think the authors have created some good work.  The presentation just needs to be improved so that the algorithm is clearer to the reader so that they can see how to reproduce the results and verify that the method will work for their application.

Comments on the Quality of English Language

The language seems to be sufficiently good for publication.

Author Response

Dear Reviewers and Editors:

We appreciate your concise and helpful review of our manuscript. We have carefully considered all of your comments and suggestions and responded to them appropriately and incorporated them in the revised manuscript.

Below is a detailed list of your comments and suggestions and our responses for your quick reference.

Notes about this response document:

  1. The reviewers’ questions/comments were copied and pasted for quick reference, and they are shown in red italics.
  2. The copied revised manuscript is shown in blue italics.
  3. Questions and suggested responses are shown in black.

Note about the revised manuscript:

  1. The modifications of the text are in review mode.
  2. The modifications of figures, tables, and references are not in review mode. The legends of the modified figures and tables and the added description of other researches are shown in red.

Yongxiang Yao (yaoyongxiang@whu.edu.cn)

Comments and Suggestions for Reviewer #1:

The paper presents a new method for registration of images collected with different sensor types to map the points between the different images. A new method using the Gaussian image pyramid using phase exponents of differences from the Gaussian image pyramid was used to find the corresponding points.  Results are presented showing that the proposed method does give better results than the previously proposed methods for a set of images.

Overall, I think that this is a high quality paper but I do have a few issues with the presentation that I think should be corrected prior to the final submission.

  1. -On page 5, the functions and elements are used in the equations without any proper definitions. Many of the times can be found through context but it is standard to define all items in equations in academic papers. For example, the item p_{s,o} (x,y) is not defined.  The functions Lodd(x,y,s,o) and Leven(x,y,s,o) are referred to in the text but not used in the following equations.  The operator \ocross needs to defined as well.  It seems to refer to convolution but this should be clearly defined somewhere.

Response:

Thanks to the comments and suggestions for the reviewer, we correct our equations and add more explicit explanation for them.

“The PC structure map is calculated as follows. First, the image I(x, y) is convolved with the odd-symmetric wavelet Lodd(x, y, s, o) and even-symmetric wavelet Leven(x, y, s, o) to obtain the response components Eso(x, y) and Oso(x, y).”

,

(1)

“where s and o denote the scale and orientation of Log-Gabor wavelets, respectively;  represents the even-symmetric filter of the Log-Gabor filter;  represents the odd-symmetric filter; i-th represents the imaginary unit of the retest; I(x, y) is the image;  represents the response result of the image on the real part filter;  represents the response result of the image on the imaginary part filter; and  represents the convolution operation.”

  1. -Equation 3 uses a PC structure map, M(x,y) and Gaussian function G(u,v,\sigma) but these are never explicitly defined in the paper.

Response:

Thanks to the comments and suggestions for the reviewer, we correct our cite to equation and add the lost equation G.

First, the scale space L(x, y, σ) is constructed from the PC structure map PC(x, y) and the Gaussian function G(u, v, σ)with the formula shown in (3), the Gaussian function G(u, v, σ) shown in (4):

,

(3)

,

(4)

  1. -On page 6, it is said that a pixel is compared with 27 points in its neighbourhood. What are these 27 points? How are they decided?

Response:

Thanks to the comments and suggestions for the reviewer, the neighbourbood points should be 26, we have changed it in our article. The reason of 26 points in SIFT explained as following:

Each pixel is compared with 8 neighboring pixels in the same layer and 9 pixels in the adjacent upper and lower layers, and the maximum and minimum values obtained are saved as extreme values. As shown in the figure below:

  1. -On page 7 and page 8, Gradient order amplitude functions are used but these have not been explicitly defined.

Response:

Thanks to the comments and suggestions for the reviewer, we add the definition of gradient order amplitude function.

,

(8)

,

(9)

,

(10)

In Equation (8~10), denotes the new second-order gradient amplitude; denotes the new second-order gradient direction; denotes the new third-order gradient amplitude; denotes the new third-order gradient direction;  and  are the gradient order amplitude function, where denotes the X-direction Sobel operator template; denotes the Y-direction Sobel operator template; σ denotes the image scale.

  1. -On pages 11-13, some explanation should be given as to what Figure 7 is actually showing. What do the green lines represent? What are the 4 images in each sub-figure? It is not to hard to figure out from context but the reader should be given some guidance.

Response:

Thanks to the comments and suggestions for the reviewer, we add more guidance in the description of Figure 7.

 Figure 7 shows the matching results of the six matching algorithms on the three types of data; the green line in the figure represents a correct match for the homonymous points, and the red line indicates a failed match. The optical-optical data shows two sets of optical image pairs of different years; the four images shown in the infra-red-optical data, where the image on the top left is an infrared image, the image on the bottom left is a thermal infrared image, and the matching images on their right are all optical images; in the map-optical data, the two on the left are electronic navigation maps, and the two on the right are the corresponding optical images. As shown in Figure 7, the proposed PEDoG algorithm can obtain the most abundant matching key-points among the three modalities and exhibits good scale and rotation invariance. In contrast, the SIFT algorithm and PSO-SIFT algorithm perform well with multi-temporal images and electronic navigation maps but have only moderate performance with infrared images. The RIFT2 algorithm is an improved version of RIFT, demonstrating good rotation invariance but not ideal scale adaptation. The OSS algorithm employs local self-similarity descriptors and performs well with both multi-temporal images and infrared images but yields fewer matching keypoints. Lastly, the HOWP algorithm uses weighted phase-oriented features for description and per-forms well, although it is sensitive to significant rotational differences.

  1. -On page 16, the outlier rejection threshold C_t is shown to have a major effect on performance but this value is not seen in any of the prior equations or descriptions of the algorithm. What exactly is this value and why does it effect the performance?

Response:

Thanks to the comments and suggestions for the reviewer, the outlier rejection threshold was a translation problem that led to a mischaracterization, and we have corrected it to feature point extraction threshold. Also, we correct the inconsistency in the description of equation. Feature Point Extraction Threshold (Ct) which we use in Eq. 7 is as following:

,

(7)

 “To comprehensively evaluate the performance of the PEDoG algorithm, an analysis of its two core parameters was conducted: the number of layers in the Difference of Gaussian (DOG) pyramid, denoted as Dl, and the feature point extraction threshold .”

  1. -More explanation is needed about the selection of the images used to test the new method. What evidence is there that the test images will have the characteristics of actual images that practitioners are likely to encounter in field applications of the algorithm? Are these images from standard test sets for this type of algorithm.  Some assurance should be given to the reader that the image selection was based on characteristics from real sensor image collection and not a selection of images that would favour the proposed algorithm. I believe that the authors actually did pick a fair data set but the paper should give some explanation to back this assertion.

Response:

Thanks to the comments and suggestions for the reviewer. For the use of experimental data, we have added the following instructions:

  1. some of the images in the experiment are from HOWP, which has been proved to be workable
  2. the remaining images were acquired with reference to the standards in HOWP.
  3. all the data in the experiment are collected in the real world, however, there are some images with larger rotation scale differences in the actual images, because our experiment fully considers the actual situation, and we try to consider the images with larger rotation differences as much as possible to validate the experiment, so as to be closer to the actual situation.

Reviewer 2 Report

Comments and Suggestions for Authors

Authors, thank you for getting the english presentation in good order.  Just a few changes though with the superscript of equation notation that is in-line with text. Remove the superscript.  Also please defin acronyms before the first use.  see attached file for corrections.

Author Response

Dear Reviewers and Editors:

We appreciate your concise and helpful review of our manuscript. We have carefully considered all of your comments and suggestions and responded to them appropriately and incorporated them in the revised manuscript.

Below is a detailed list of your comments and suggestions and our responses for your quick reference.

Notes about this response document:

  1. The reviewers’ questions/comments were copied and pasted for quick reference, and they are shown in red italics.
  2. The copied revised manuscript is shown in blue italics.
  3. Questions and suggested responses are shown in black.

Note about the revised manuscript:

  1. The modifications of the text are in review mode.
  2. The modifications of figures, tables, and references are not in review mode. The legends of the modified figures and tables and the added description of other researches are shown in red.

Yongxiang Yao (yaoyongxiang@whu.edu.cn)

Comments and Suggestions for Reviewer #2:

Authors, thank you for getting the english presentation in good order.  Just a few changes though with the superscript of equation notation that is in-line with text. Remove the superscript.  Also please defin acronyms before the first use.  see attached file for corrections.

Response:

Thanks to the comments and suggestions for the reviewer, we have removed superscript of equation in in-line text. We have also defined the acronyms when it is first used.

The acronyms were defined as following:

“…and the results show that the NCM (number of correct matches) of the PEDoG method…

…Conventional feature-based image registration methods like SIFT (the scale invariant feature transform) [5] and SURF (speeded-up robust features) [6] typically rely on gradient information to extract and describe features…

…These include techniques such as directional gradient distance histograms combined with grey wolf optimization [11], Log-Gabor filter-optimized matching HOPC (histogram of orientated phase congruency) [12], and CFOG (channel features of orientated gradients) [13] ….

…Ma et al. introduced the PSO-SIFT (position scale orientation - SIFT) algorithm [15], which performs well in handling nonlinear brightness differences and rotation changes through the establishment of new image gradient features, but it remains sensitive to contrast differences and signal-to-noise ratio variations. Sedaghat et al. proposed the HOSS (histogram of oriented self-similarity) algorithm [36] based on self-similarity features, which ensures good rotational invariance but performs poorly in multi-modal images with high contrast and nonlinear radiometric distortions. The RRSS (rank-based ratio self-similarity) method [37] proposed by Xiong et al. effectively addresses differences between multi-modal images but incurs some loss in rotational invariance. …

… The Log-Polar Histogram framework is a classical framework that has been success-fully applied in multi-modal image matching. The Log-Polar descriptor approach using the Gradient Location Orientation Histogram (GLOH) algorithm is notably advantageous and stable …”

The superscript was removed like the following:

“… in Equation (6), denotes the phase-exponential Gaussian difference pyramid result. In order to better demonstrate the results of the phase-exponential Gaussian difference pyramid

… where  denotes the normalized image; denotes the original image; (x, y) denotes the coordinates of the image; is a local window centered on (x, y) with a size of (2×s+1)×(2×s+1); and denotes the taking of an absolute value.

In Equation (9~11), denotes the new second-order gradient amplitude; denotes the new second-order gradient direction; denotes the new third-order gradient amplitude; denotes the new third-order gradient direction;  and  are the gradient order amplitude function, where denotes the X-direction Sobel operator template; denotes the Y-direction Sobel operator template; σ denotes the image scale.

where N is the number of homonymous points,  is the coordinate of the i-th truth point  after the corresponding matching transformation.

 Detailed pictures are attached.

Reviewer 3 Report

Comments and Suggestions for Authors

In order to improve the matching geometric invariance and accuracy, this manuscript introduces phase consistency model into the differential Gaussian pyramid to construct a new phase index. However, some concerns can be addressed before further review process.

1)      In the 17th line, author’s method could use present tense, not past tense.

2)      In the 21th line, it has three types of MMI.

3)      In the 23th line, ‘NCM’ first appeared in abstract. In general, the first occurrence of an abbreviation should be followed by its full name. Moreover, abbreviations in articles should be defined first and then used, such as SIFT in the 107th and 72th lines.

4)      The first letter of the keyword part could be capitalized.

5)      In the 25th line, unlike PSNR, RMSE does not have a unit of measurement. RMSE is the measure of the average error between predicted values and actual observations, not associated with any specific unit.

6)      When introducing the work of others in the article, you could cite it in the references and add an index of the citation in the cited section, such as the 114th-118th lines. Authors are advised to check the full paper.

7)      Figure annotations should be centered consistently throughout the paper.

8)      Letters in Equation should be enclosed by $$ in latex, such as Equation (1) and the lines from the 191th to 196th.

9)      All equations could be followed by a comma or period.

10)  In the 324th line, ‘Correct Match Number’ and ‘NCM’ are a unified expression.

11)  In the paper, the indexing of Equations or figures should be consistent, e.g.Fig.3 and Figure 5, Equation 10 and Eqs. (8-9).

12)  In the 283th and 285th lines, GLOH’ is defined twice and different every time.

13)  There are some incorrect references in the 336th, such as ‘Figure x(a)’.

Author Response

Dear Reviewers and Editors:

We appreciate your concise and helpful review of our manuscript. We have carefully considered all of your comments and suggestions and responded to them appropriately and incorporated them in the revised manuscript.

Below is a detailed list of your comments and suggestions and our responses for your quick reference.

Notes about this response document:

  1. The reviewers’ questions/comments were copied and pasted for quick reference, and they are shown in red italics.
  2. The copied revised manuscript is shown in blue italics.
  3. Questions and suggested responses are shown in black.

Note about the revised manuscript:

  1. The modifications of the text are in review mode.
  2. The modifications of figures, tables, and references are not in review mode. The legends of the modified figures and tables and the added description of other researches are shown in red.

Yongxiang Yao (yaoyongxiang@whu.edu.cn)

Comments and Suggestions for Reviewer #3:

In order to improve the matching geometric invariance and accuracy, this manuscript introduces phase consistency model into the differential Gaussian pyramid to construct a new phase index. However, some concerns can be addressed before further review process.

  1. -In the 17th line, author’s method could use present tense, not past tense.

Response:

Thank you for your suggestion, we have changed the tense from past to present.

“Based on this, a lightweight MMI alignment of phase exponent of difference of gaussian-pyramid (PEDoG) was is proposed that take into account phase exponent of difference of gaussian-pyramid with normalized filtration”

  1. -In the 21th line, it has three types of MMI.

Response:

Thank you for your suggestion, we correct the number of MMI.

“Then four three types of MMI (multi-temporal image, infrared-optical image, map-optical image) are selected as the experimental dataset”

  1. - In the 23th line, ‘NCM’ first appeared in abstract. In general, the first occurrence of an abbreviation should be followed by its full name. Moreover, abbreviations in articles should be defined first and then used, such as SIFT in the 107th and 72th lines.

Response:

Thank you for your suggestion, we add the definition for these abbreviations in article.

“…and the results show that the NCM (number of correct matches) of the PEDoG method…

…Conventional feature-based image registration methods like SIFT (the scale invariant feature transform) [5] and SURF (speeded-up robust features) [6] typically rely on gradient information to extract and describe features…

  1. - The first letter of the keyword part could be capitalized.

Response:

Thanks to the comments and suggestions from the reviewer, we capitalized the first letter of the keyword.

“Keywords: Multi-modal images; Phase exponent of difference of gaussian-pyramid; Normalized filtering; Log-polar coordinate descriptor; Image registration”

  1. - In the 25th line, unlike PSNR, RMSE does not have a unit of measurement. RMSE is the measure of the average error between predicted values and actual observations, not associated with any specific unit.

Response:

Thanks to the comments and suggestions from the reviewer, we disassociate the RMSE with the specific unit.

“… the average RMSE is 1.69 …”

  1. - When introducing the work of others in the article, you could cite it in the references and add an index of the citation in the cited section, such as the 114th-118th lines. Authors are advised to check the full paper.

Response:

Thanks to the comments and suggestions from the reviewer, we cite the articles in the references and add the index of the citation in the cited section.

“Sedaghat et al. proposed the HOSS (histogram of oriented self-similarity) algorithm [36] based on self-similarity features, which ensures good rotational invariance but performs poorly in multi-modal images with high contrast and nonlinear radiometric distortions. The RRSS (rank-based ratio self-similarity) method [37] proposed by Xiong et al. effectively addresses differences between multi-modal images but incurs some loss in rotational invariance.”

“[36] (HOSS)Sedaghat A, Mohammadi N. Illumination-robust remote sensing image matching based on oriented self-similarity[J]. ISPRS Journal of Photogrammetry and Remote Sensing, 2019, 153: 21-35.

[37] (RRSS)Xiong X, Jin G, Xu Q, et al. Robust SAR image registration using rank-based ratio self-similarity[J]. IEEE Journal of Selected Topics in Applied Earth Observations and Remote Sensing, 2021, 14: 2358-2368.”

  1. - Figure annotations should be centered consistently throughout the paper

Response:

Thanks to the comments and suggestions from the reviewer, we center the annotations of our figure.

  1. - Letters in Equation should be enclosed by $$ in latex, such as Equation (1) and the lines from the 191th to 196th.

Response:

Thanks to the comments and suggestions from the reviewer, we correct formatting of formulas in the text.

“First, the image I(x, y) is convolved with the odd-symmetric wavelet Lodd(x, y, s, o) and even-symmetric wavelet Leven(x, y, s, o) to obtain the response components Eso(x, y) and Oso(x, y).”

“where s and o denote the scale and orientation of Log-Gabor wavelets, respectively;  represents the even-symmetric filter of the Log-Gabor filter;  represents the odd-symmetric filter; i-th represents the imaginary unit of the retest; I(x, y) is the image;  represents the response result of the image on the real part filter;  represents the response result of the image on the imaginary part filter; and  represents the convolution operation.

  1. - All equations could be followed by a comma or period.

Response:

Thanks to the comments and suggestions from the reviewer, we add a comma after each equation.

  1. - In the 324th line, ‘Correct Match Number’ and ‘NCM’ are a unified expression.

Response:

Thanks to the comments and suggestions from the reviewer, we correct the definition for the abbreviations in article.

“…the results show that the NCM (number of correct matches) of the PEDoG method is a minimum of 3.3 times improvement compared with the other methods, and the average RMSE (Root Mean Square Error) is 1.69 …”

“The quantitative evaluation employs three metrics to assess the matching performance of the methods: Success Rate (SR), NCM, and RMSE.”

  1. - In the paper, the indexing of Equations or figures should be consistent, e.g.Fig.3 and Figure 5, Equation 10 and Eqs. (8-9).

Response:

Thanks to the comments and suggestions from the reviewer, we change the index to be consistent. Figure for figure and Equation for equation.

“As can be seen from Figure 3, the feature points are basically distributed over the image structure and a large number of corresponding feature points are detected in multimodal remote sensing image pairs.”

“In Equation (9~11), denotes the new second-order gradient amplitude; denotes the new second-order gradient direction; denotes the new third-order gradient amplitude; denotes the new third-order gradient direction;  and  are the gradient order amplitude function, where denotes the X-direction Sobel operator template; denotes the Y-direction Sobel operator template; σ denotes the image scale.

  1. - In the 283th and 285th lines, ‘GLOH’ is defined twice and different every time.

Response:

Thanks to the comments and suggestions from the reviewer, we unify the definition for GLOH.

“The Log-Polar Histogram framework is a classical framework that has been success-fully applied in multi-modal image matching. The Log-Polar descriptor approach using the Gradient Location Orientation Histogram (GLOH) algorithm is notably advantageous and stable”

  1. - There are some incorrect references in the 336th, such as ‘Figure x(a)’.

Response:

Thanks to the comments and suggestions from the reviewer, we correct the error as ‘Figure 6(a)’.

Detailed pictures are attached.

Reviewer 4 Report

Comments and Suggestions for Authors

1. The literature research is not sufficient, and it is necessary to add some literature to provide a detailed explanation of the current research status at home and abroad.

2. The formula in the manuscript should be as flat as the text.

3. Language needs polishing, MDPI's polishing service can be used.

4.Why choose these datasets in the experimental datasets section?Whether the number of  image pairs is enough or not?The image sizes ranged from 381 pixels to 750 pixels, if not in the range, is the algorithm feasible?

5. In section of analysis of scale-invariance, the reference images ranges from 0.6x to 1.8x, if less than 0.6x or more than1.8x,is the matching results feasible?

6. The reference image was rotated in both clockwise and counterclockwise directions at 30-degree intervals,why is 30-degree intervals?

Comments on the Quality of English Language

Language needs polishing, MDPI's polishing service can be used.

Author Response

Dear Reviewers and Editors:

We appreciate your concise and helpful review of our manuscript. We have carefully considered all of your comments and suggestions and responded to them appropriately and incorporated them in the revised manuscript.

Below is a detailed list of your comments and suggestions and our responses for your quick reference.

Notes about this response document:

  1. The reviewers’ questions/comments were copied and pasted for quick reference, and they are shown in red italics.
  2. The copied revised manuscript is shown in blue italics.
  3. Questions and suggested responses are shown in black.

Note about the revised manuscript:

  1. The modifications of the text are in review mode.
  2. The modifications of figures, tables, and references are not in review mode. The legends of the modified figures and tables and the added description of other researches are shown in red.

Yongxiang Yao (yaoyongxiang@whu.edu.cn)

Comments and Suggestions for Reviewer #4:

  1. -The literature research is not sufficient, and it is necessary to add some literature to provide a detailed explanation of the current research status at home and abroad.

Response:

Thank you for your suggestions. We have extensively reviewed relevant papers and supplemented the references. Additionally, we have enhanced the "Related Work" section, providing a detailed explanation of the current research status on multi-modal remote sensing image matching both internationally and domestically.

Region-based registration methods establish registration relationships between image pairs using the raw pixel values and specific similarity metrics. These methods can be broadly categorized into three classes, including correlation-based methods [8], Fourier-based methods [9], and mutual information-based methods [10]. In recent years, some researchers have introduced phase correlation techniques, leading to rapid advancements in multi-modal image matching. Building upon this, experts and scholars have developed methods that combine phase congruency models, self-similarity features, and enhanced gradient features. These include techniques such as directional gradient distance histograms combined with grey wolf optimization [11], Log-Gabor filter-optimized matching HOPC (histogram of orientated phase congruency) [12],  CFOG (channel features of orientated gradients) [13], Angle-Weighted Orientation Gradient (AWOG)[43] and Multi-Orientation Tensor Index Feature (MoTIF)[41]. These approaches effectively overcome nonlinear radiometric distortions and contrast differences between multi-modal images and exhibit strong robustness in handling dis-placement variations in images. However, it's worth noting that such methods primarily focus on translational shifts, and if images involve complex geometric transformations, registration methods may encounter challenges.

Feature-based methods in the field of image registration began with Lowe et al.'s introduction of Scale Invariant Feature Transform (SIFT) matching [5], leading to the rapid development of various SIFT-like techniques [14]. However, gradient features are unable to adapt to the modality differences in multi-modal images, making such methods unsuitable for multi-modal image matching. Ma et al. introduced the PSO-SIFT (position scale orientation - SIFT) algorithm [15], which performs well in handling nonlinear brightness differences and rotation changes through the establishment of new image gradient features, but it remains sensitive to contrast differences and signal-to-noise ratio variations. Sedaghat et al. proposed the HOSS (histogram of oriented self-similarity) algorithm [36] based on self-similarity features, which ensures good rotational invariance but performs poorly in multi-modal images with high contrast and nonlinear radiometric distortions. The RRSS (rank-based ratio self-similarity) method [37] proposed by Xiong et al. effectively addresses differences between multi-modal images but incurs some loss in rotational invariance.

Some experts and researchers have tackled multi-modal image matching from the perspective of phase congruency models. For example, Li et al. introduced a radiometrically invariant transform feature matching method [18], which uses a maximum index map to overcome NRD differences in MMRSI effectively. However, it relies on a strategy involving circular feature calculations to overcome rotational differences, which results in lower technical efficiency. Xiang et al. [40] enhanced the PC model, keypoint extraction, and similarity measurement methods, constructing features for optical and SAR image matching. Fan et al. [39] designed a multi-scale PC descriptor known as Multi-Scale Adaptive Block Phase Congruency (MABPC). This descriptor utilizes multi-scale phase congruency features encoded with an adaptive block spatial structure. Yao et al. proposed absolute phase orientation histogram matching, designing absolute phase-oriented features to accommodate differences between multi-modal images and resist scale, displacement, and rotational differences. However, this method is limited to matching tasks with small rotational differences. Yang et al. introduced local phase sharpness-oriented features to adapt to MMRSI matching and improve the applicability to rotational differences in multi-modal images but still lacks complete local rotational invariance.

Recently, various algorithms have been proposed to address multi-modal image differences by improving image scale space, such as the CoFSM algorithm [21], which reduces multi-modal image differences by improving the image scale space. Other approaches include the MS-HLMO algorithm [22], which is based on multiscale joint mean minimal gradient features, and multi-modal image matching through local normalized image filtering [23]. These methods have improved the matching performance of multi-modal remote sensing images, demonstrating strong rotational invariance. However, they still suffer from issues such as low accuracy of corresponding points and poor matching robustness to varying degrees.

With the rapid development of artificial intelligence theory, deep learning techniques have been introduced into multi-modal image matching. Examples include convolutional neural network-based matching [24] and graph neural network-based matching (SuperGlue) [25]. These methods have demonstrated excellent performance in homogenous image matching but have shown limitations when applied to multi-modal images. In response to this challenge, researchers have explored methods specifically designed for multi-modal image matching. These include the D2-Net network for multi-source image feature extraction and description [26], the LoFTR algorithm based on transformer networks [27], and its improved version SE2-LoFTR with rotational invariance [28]. M2DT-Net [42] is a method that combines learned features with Delaunay triangulation constraints, but it has certain limitations in handling radiometric invariance. Deep learning-based approaches offer speed and strong feature learning capabilities, especially in the context of multi-modal image matching, where they have shown significant potential. However, due to substantial variations in land features between multi-modal images and the difficulty in obtaining training samples, the generalization and applicability of such methods are currently limited.

The added references are as follows:

  1. Sedaghat A, Mohammadi N. Illumination-robust remote sensing image matching based on oriented self-similarity[J]. ISPRS Journal of Photogrammetry and Remote Sensing, 2019, 153: 21-35.
  2. Xiong X, Jin G, Xu Q, et al. Robust SAR image registration using rank-based ratio self-similarity[J]. IEEE Journal of Selected Topics in Applied Earth Observations and Remote Sensing, 2021, 14: 2358-2368.
  3. Xiong X, Jin G, Xu Q, et al. Self-similarity features for multimodal remote sensing image matching[J]. IEEE Journal of Selected Topics in Applied Earth Observations and Remote Sensing, 2021, 14: 12440-12454.
  4. Fan, J.; Ye, Y.; Li, J.; Liu, G.; Li, Y. A Novel Multiscale Adaptive Binning Phase Congruency Feature for SAR and Optical Image Registration. IEEE Trans. Geosci. Remote Sens. 2022, 60, 1-16.
  5. Xiang, Y.; Tao, R.; Wang, F.; You, H.; Han, B. Automatic Registration of Optical and SAR Images Via Improved Phase Congruency Model. IEEE Journal of Selected Topics in Applied Earth Observations and Remote Sensing 2020, 13, 5847-5861.
  6. Yao, Y.; Zhang, B.; Wan, Y.; Zhang, Y. Motif: multi-orientation tensor index feature descriptor for sar-optical image registration. Int. Arch. Photogramm. Remote Sens. Spatial Inf. Sci. 2022, XLIII-B2-2022, 99–105.
  7. Zhang, Y.; Liu, Y.; Zhang, H.; Ma, G. Multimodal Remote Sensing Image Matching Combining Learning Features and Delaunay Triangulation. IEEE Trans. Geosci. Remote. Sens. 2022, 60, 1-17.
  8. Fan, Z.; Zhang, L.; Liu, Y.; Wang, Q.; Zlatanova, S. Exploiting High Geopositioning Accuracy of SAR Data to Obtain Accurate Geometric Orientation of Optical Satellite Images. Remote Sens. 2021, 13, 3535.
  9. - The formula in the manuscript should be as flat as the text.

Response:

We offer the following improvements and supplements in response to the above comment. We have changed the formulas in the manuscript to make sure they are as flat as the text.

  1. - Language needs polishing, MDPI's polishing service can be used.

Response:

Thank you for your valuable feedback. We have touched up the language of the manuscript, and we apologize for not using MDPI's touch-up service due to time constraints.

  1. - Why choose these datasets in the experimental datasets section?Whether the number of image pairs is enough or not?The image sizes ranged from 381 pixels to 750 pixels, if not in the range, is the algorithm feasible?

Response:

We offer the improvements and supplements in response to the above comment.

The selection of these datasets in the experimental dataset section is based on the fact that this multi-modal remote sensing image dataset almost covers all application scenarios of multi-modal image matching, including multi-source data interpretation, multi-structural data registration, and multi-spectral data fusion. Due to differences in temporal phases, lighting conditions, and sensors, there are significant scale variations, rotation differences, translation differences, and radiometric differences between image pairs. Therefore, this dataset is representative and can be used to validate and compare the performance of multi-modal remote sensing image registration algorithms.

Part of the experimental dataset is sourced from the HOWP paper, which has been validated to showcase the characteristics of multi-modal images.

This study involves 60 pairs of experimental image pairs, and in the process of comparison with algorithms such as HOWP and RIFT, we find that the number of image pairs is quite sufficient to meet the matching requirements. This is enough to fully demonstrate the performance of the proposed algorithm.

The proposed algorithm is adaptable to image matching at different scales. To verify the algorithm's performance in matching images at different scales, we conducted down-sampling and up-sampling simulations on six sets of images from the manuscript. The matching results are as follows:

Detailed pictures are attached.

In summary, our algorithm demonstrates excellent matching results at different scales, validating its robustness and resilience to scale variations. This further confirms the effectiveness of our proposed algorithm in handling image matching at different scales.

References for HOWP and RIFT:

Zhang, Y., Yao, Y., Wan, Y., Liu, W., Yang, W., Zheng, Z., & Xiao, R. Histogram of the orientation of the weighted phase descriptor for multi-modal remote sensing image matching. ISPRS Journal of Photogrammetry and Remote Sensing, 2023,196, 1-15.

Li, J., Hu, Q., & Ai, M. (2019). RIFT: Multi-modal image matching based on radiation-variation insensitive feature transform. IEEE Transactions on Image Processing, 29, 3296-3310.

  1. - In section of analysis of scale-invariance, the reference images ranges from 0.6x to 1.8x, if less than 0.6x or more than1.8x,is the matching results feasible?

Response:

We offer the improvements and supplements in response to the above comment.

Our algorithm demonstrates robust scale invariance, extending beyond the range of 0.6x to 1.8x. It is capable of handling image pairs with even more extensive scale differences. To further test the matching performance, we conducted additional experiments by scaling down and up the images. The obtained matching results are presented below:

The selection of the scale difference range from 0.6x to 1.8x was primarily inspired by the testing approach outlined in the HOWP paper. In satellite or unmanned aerial vehicle (UAV) image matching, when there is significant image disparity, coarse correction can be performed using geographic coordinates such as RPC (Rational Polynomial Coefficients) before matching. From the observed results above, it is evident that with increasing scale differences, the number of obtained normalized cross-correlation maps (NCMs) gradually decreases but remains sufficient for the requirements of the registration operation. In summary, the PEDoG algorithm exhibits scale invariance.

  1. - The reference image was rotated in both clockwise and counterclockwise directions at 30-degree intervals,why is 30-degree intervals?

Response:

We offer the following improvements and supplements in response to the above comment.

The algorithm presented in this paper exhibits a 360-degree rotational invariance, as tested through rotation-invariant matching using the same images mentioned in the text. Initially, the reference image was rotated at intervals of 10 degrees in both clockwise and counterclockwise directions, generating a total of 36 simulated images in 18 directions. The matching results are illustrated in the figure below ("- " denotes counterclockwise rotation).

The reference image is rotated in both clockwise and counterclockwise directions at intervals of 30 degrees. This is done to save manuscript space, and a 30-degree rotation interval is commonly used in rotation tests. With rotation differences set within [-180°, 180°], the algorithm consistently achieves successful matches and produces rich normalized cross-correlation maps (NCM). In summary, the proposed algorithm demonstrates rotational invariance. Compared to traditional methods, the proposed algorithm exhibits robust matching performance in multi-modal remote sensing image (MRSI) matching, particularly in handling nonlinear radiometric distortions, scale variations, and rotational invariance.

Detailed pictures are attached.

Reviewer 5 Report

Comments and Suggestions for Authors

In multi-modal imagery, differences in imaging mechanisms frequently result in large disparities in signal-to-noise ratios (SNR) between modalities. As a result, matching algorithms face a challenge in maintaining both matching geometric invariance and accuracy. As a result, attaining high-precision matching correspondences while minimising SNR interference in multi-modal remote sensing image matching is difficult. To solve this issue, this interesting study provides a lightweight multi-modal image registration technique that uses phase-indexed difference pyramids and normalised filtering to achieve high-precision matching correspondences while preserving geometric invariance. 

The article overall seems well written and worthy of publication. The references are also up-to-date, illustrating the state of the art in research.

Asking for more detail in the description of the methods and presentation of the results, I suggest a re-reading to some inaccuracies: 

- Line 181, 221, 240, 242, 343, 344, 413, 419, 443, 445, 452: all figures must be mentioned in the main text as Figure 1, Figure 2, etc. before they are included. 

- Line 221, 242, 344, 388, 402, 484: all figures must be mentioned in the main text as Figure 1, Figure 2, etc., without abbreviating in Fig. 1, Fig. 2 etc.

- Line 336: could be 5(a) instead?

- Line 338: could be 5(b) instead?

Author Response

Dear Reviewers and Editors:

We appreciate your concise and helpful review of our manuscript. We have carefully considered all of your comments and suggestions and responded to them appropriately and incorporated them in the revised manuscript.

Below is a detailed list of your comments and suggestions and our responses for your quick reference.

Notes about this response document:

  1. The reviewers’ questions/comments were copied and pasted for quick reference, and they are shown in red italics.
  2. The copied revised manuscript is shown in blue italics.
  3. Questions and suggested responses are shown in black.

Note about the revised manuscript:

  1. The modifications of the text are in review mode.
  2. The modifications of figures, tables, and references are not in review mode. The legends of the modified figures and tables and the added description of other researches are shown in red.

Yongxiang Yao (yaoyongxiang@whu.edu.cn)

Comments and Suggestions for Reviewer #5:

In multi-modal imagery, differences in imaging mechanisms frequently result in large disparities in signal-to-noise ratios (SNR) between modalities. As a result, matching algorithms face a challenge in maintaining both matching geometric invariance and accuracy. As a result, attaining high-precision matching correspondences while minimising SNR interference in multi-modal remote sensing image matching is difficult. To solve this issue, this interesting study provides a lightweight multi-modal image registration technique that uses phase-indexed difference pyramids and normalised filtering to achieve high-precision matching correspondences while preserving geometric invariance.

The article overall seems well written and worthy of publication. The references are also up-to-date, illustrating the state of the art in research.

Asking for more detail in the description of the methods and presentation of the results, I suggest a re-reading to some inaccuracies:

  1. - Line 181, 221, 240, 242, 343, 344, 413, 419, 443, 445, 452: all figures must be mentioned in the main text as Figure 1, Figure 2, etc. before they are included.

Response:

Thanks to the comments and suggestions from the reviewer, we add the mention of each figure when they are included.

  1. - Line 221, 242, 344, 388, 402, 484: all figures must be mentioned in the main text as Figure 1, Figure 2, etc., without abbreviating in Fig. 1, Fig. 2 etc.

Response:

Thanks to the comments and suggestions from the reviewer, we unify the mention as Figure 1 rather than Fig 1.

“… Following in Figure 1, shows the overall flow chart of the proposed method. …

… a set of infrared-optical images are selected to be significant in this paper, and the results are shown in Figure 2. …

… As can be seen from Figure 3, the feature points are basically distributed over the image structure and a large number of corresponding feature points are detected in multimodal remote sensing image pairs. …”

  1. - Line 336: could be 5(a) instead?

Response:

Thanks to the comments and suggestions from the reviewer, it should be Figure 6(a).

“In Figure 6(a), "Image Type 1" represents multi-temporal optical-optical, "Image Type 2" represents infrared-optical, and "Image Type 3" represents map-optical.”

  1. - Line 338: could be 5(b) instead?

Response:

Thanks to the comments and suggestions from the reviewer, it should be Figure 6(b).

“In Figure 6(b), the symbol "+" indicates matching failures or RMSE 7 pixels. The units for SR are percentage (%), NCM is in point numbers, and RMSE is in pixels.”

Round 2

Reviewer 1 Report

Comments and Suggestions for Authors

The authors have answered all of my comments/questions about the previous draft of the paper to my satisfaction. I only recommend that they give the paper another proofread to check for minor grammar and other language issues.  For example, the section "Relate Work" should probably be "Related Work".

Comments on the Quality of English Language

There are some minor language issues that need to be corrected.

Reviewer 3 Report

Comments and Suggestions for Authors

There are still some editing errors.

1) Why are (a) and (b) of Figure 5 plus semicolons?

2) In the Abstract, the RMSE (Root Mean Square Error) abbreviation with others is not unified. The first letters of others are not capital cases.

Comments on the Quality of English Language

Recheck the whole paper.

Reviewer 4 Report

Comments and Suggestions for Authors

The manuscript can be accepted in present form.